# Sequence-specific capture and concentration of viral RNA by type III CRISPR system enhances diagnostic

Anna Nemudraia[1,3], Artem Nemudryi [1,3], Murat Buyukyoruk[1], Andrew M. Scherffius[1], Trevor Zahl[1], Tanner Wiegand [1], Shishir Pandey[1], Joseph E. Nichols[1], Laina N. Hall[1], Aidan McVey[1], Helen H. Lee[1], Royce A. Wilkinson [1], Laura R. Snyder[2], Joshua D. Jones [2], Kristin S. Koutmou [2], Andrew Santiago-Frangos [1] ✉ & Blake Wiedenheft [1] ✉

Type-III CRISPR-Cas systems have recently been adopted for sequence-specific detection of SARS-CoV-2. Here, we repurpose the type III-A CRISPR complex from *Thermus thermophilus* (TtCsm) for programmable capture and concentration of specific RNAs from complex mixtures. The target bound TtCsm complex generates two cyclic oligoadenylates (i.e., $cA_3$ and $cA_4$) that allosterically activate ancillary nucleases. We show that both Can1 and Can2 nucleases cleave single-stranded RNA, single-stranded DNA, and double-stranded DNA in the presence of $cA_4$. We integrate the Can2 nuclease with type III-A RNA capture and concentration for direct detection of SARS-CoV-2 RNA in nasopharyngeal swabs with 15 fM sensitivity. Collectively, this work demonstrates how type-III CRISPR-based RNA capture and concentration simultaneously increases sensitivity, limits time to result, lowers cost of the assay, eliminates solvents used for RNA extraction, and reduces sample handling.

Although qPCR (quantitative polymerase chain reaction) remains the "gold standard" for nucleic acid detection, it requires sophisticated equipment, trained personnel, efficient specimen transport to high-complexity labs, and reliable reporting systems[1]. While the complexity and turnaround times necessary for qPCR are acceptable for many diagnostic applications, the SARS-CoV-2 (Severe Acute Respiratory Syndrome Coronavirus 2) pandemic reveals an urgent need for diagnostics that are easy to distribute, simple to perform, and fast enough to stop transmission of a contagious disease[1]. Although rapid antigen tests and isothermal amplification methods have helped address this need, these and other emerging methods have limitations related to sensitivity, versatility, or specificity[2,3].

CRISPR RNA-guided diagnostics (CRISPR-dx) are a diverse group of nascent technologies that aim to address current limitations by providing a versatile and programmable platform that is sufficiently sensitive for clinical applications and stable enough for distribution[4,5]. The first CRISPR-based viral diagnostic came from Collins and colleagues in 2016, when they demonstrated that Cas9 could be used to discriminate between different variants of the Zika virus[6]. This approach relies on converting viral RNA to DNA using reverse transcriptase, followed by isothermal DNA amplification prior to sequence-based discrimination by Cas9. The exclusive recognition of double-stranded DNA (dsDNA) by Cas9 seemed to be an intrinsic limitation for diagnostic applications that require RNA detection. However, Beisel and colleagues recently developed a creative method that uses the trans-acting CRISPR-RNA (tracrRNA) to capture complementary RNA guides derived from RNA viruses[7]. In this system, the engineered tracrRNA-crRNA hybrid guides Cas9 to a complementary dsDNA reporter. While this approach enables RNA detection, Cas9 is a single turn-over enzyme, which may limit its sensitivity. In contrast to Cas9, target recognition by

[1]Department of Microbiology and Cell Biology, Montana State University, Bozeman, MT 59717, USA. [2]Department of Chemistry, University of Michigan, Ann Arbor, MI 48105, USA. [3]These authors contributed equally: Anna Nemudraia, Artem Nemudryi. ✉e-mail: andrew.santiagofrangos@gmail.com; bwiedenheft@gmail.com

type V (Cas12-DETECTR) and type VI (Cas13-SHERLOCK) CRISPR-systems activates a multi-turnover non-sequence-specific "collateral nuclease" activity that amplifies the signal by cleaving thousands of reporter molecules for every target bound[8,9]. The recently reported rates of collateral cleavage by target bound Cas12 and Cas13 nucleases indicate that the theoretical limit of detection, without prior amplification of the target is 1 pM (~$10^6$ copies/μL)[10,11].

Like type VI, type III CRISPR systems also recognize RNA. However, unlike any other system, target recognition by type III complexes simultaneously activates polymerase and HD-nuclease domains in the Cas10 subunit[12–14]. The polymerase domain has been estimated to generate ~1000 cyclic oligoadenylates per bound RNA[15], which trans-activate multi-turnover ancillary nucleases that provide defense from invading genetic parasites[16,17]. Thus, this biochemical cascade provides two signal amplifications steps, the second of which activates thousands of multi-turnover ancillary nucleases, rather than relying on a single collateral nuclease (i.e., Cas13) bound to a target. However, initial efforts to implement this approach failed to be sufficiently sensitive for clinical applications without prior amplification of the target RNA[18–20]. The sensitivity of this first-generation diagnostic was in part limited by the use of an ancillary nucleases (e.g., Csm6) that also degrade the cyclic nucleotide activator[21–25]. Recently, Malcolm White's lab demonstrated that alternative ancillary nucleases, which efficiently cleave reporters but do not cleave the signaling molecule, can be used to enhance the sensitivity of type III-based diagnostics[26].

Despite innovations leading to new and improved CRISPR-based diagnostics, point-of-care testing requires new strategies that simplify the workflow and increase the sensitivity without prior RNA purification or amplification (e.g., PCR, LAMP, NASB, RPA, etc.). Here, we bring CRISPR-dx closer to a deployable diagnostic by developing a type III CRISPR-based method for sequence-specific capture and concentration of a specific RNA from a complex mixture. To improve the sensitivity, we purify several different ancillary nucleases (i.e., Can1, Can2,

and NucC), systemically test nuclease activation using a series of purified cyclic oligoadenylate standards, and test for ring nuclease activity. We show that Can1 and Can2 nucleases cleave single-stranded RNA (ssRNA) and DNA (ssDNA), as well as double-stranded DNA (dsDNA) in the presence of a Cas10-generated cyclic oligomer composed of four adenosines (i.e., $cA_4$). We demonstrate how the type III complex can be used to bypass RNA extraction methods, and that coupling type III-based RNA capture with the AaCan2 nuclease further increases the sensitivity of SARS-CoV-2 RNA detection in patient swabs to 15 fM ($8.3 \times 10^3$ copies/μL).

## Results

### Type III-mediated sequence-specific enrichment of RNA

Type III CRISPR RNA-guided complexes (i.e., Csm and Cmr) bind and cleave complementary single-stranded RNA (ssRNA) targets[27]. Complementary RNA is cleaved in six-nucleotide increments by metal-dependent nucleases (Csm3 or Cmr4) that form the oligomeric "backbone" of the complex[28]. Type III complexes release fragments of the cleaved target, which inactivates ATP polymerization by the Cas10 subunit[28]. Previously, we mutated residues in the Csm3 subunit responsible for target RNA cleavage (D34A), purified the RNase-dead complex (TtCsm$^{Csm3-D34A}$), and showed that the mutant complex provides more sensitive detection of viral RNA than the wild-type complex[18]. To further increase the sensitivity, we set out to determine if TtCsm$^{Csm3-D34A}$ could be used to concentrate sequence-specific RNAs. To test this approach, we mixed $^{32}$P-labeled target or non-target RNAs with TtCsm$^{Csm3-D34A}$, incubated for 20 min, and concentrated the His-tagged complex using nickel-derivatized magnetic beads (Fig. 1a and Supplementary Fig. 1a). The beads were concentrated using a magnet, and RNAs were extracted from the bound and unbound fractions. The type III complex captured most of the radiolabeled target RNA (76 ± 5.8%), while non-target RNA primarily remains in the supernatant (Fig. 1b and Supplementary Fig. 1b, c). To determine if

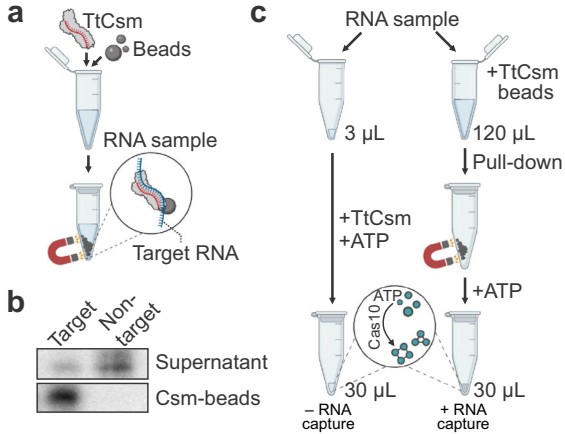

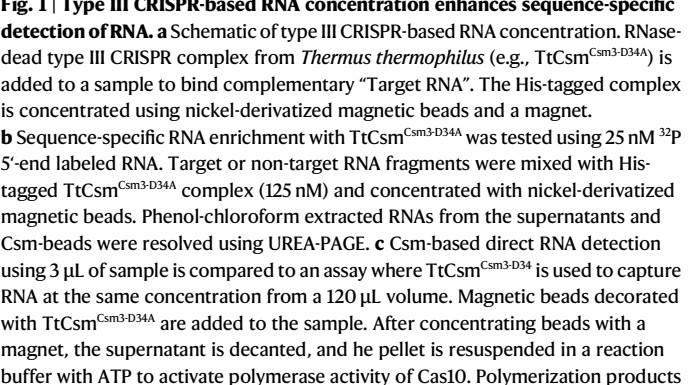

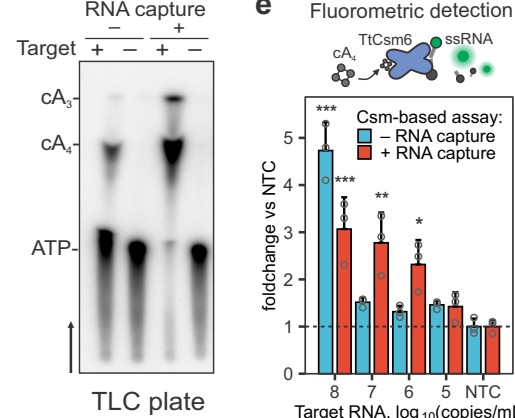

**Fig. 1 | Type III CRISPR-based RNA concentration enhances sequence-specific detection of RNA. a** Schematic of type III CRISPR-based RNA concentration. RNase-dead type III CRISPR complex from *Thermus thermophilus* (e.g., TtCsm$^{Csm3-D34A}$) is added to a sample to bind complementary "Target RNA". The His-tagged complex is concentrated using nickel-derivatized magnetic beads and a magnet. **b** Sequence-specific RNA enrichment with TtCsm$^{Csm3-D34A}$ was tested using 25 nM $^{32}$P 5'-end labeled RNA. Target or non-target RNA fragments were mixed with His-tagged TtCsm$^{Csm3-D34A}$ complex (125 nM) and concentrated with nickel-derivatized magnetic beads. Phenol-chloroform extracted RNAs from the supernatants and Csm-beads were resolved using UREA-PAGE. **c** Csm-based direct RNA detection using 3 μL of sample is compared to an assay where TtCsm$^{Csm3-D34}$ is used to capture RNA at the same concentration from a 120 μL volume. Magnetic beads decorated with TtCsm$^{Csm3-D34A}$ are added to the sample. After concentrating beads with a magnet, the supernatant is decanted, and he pellet is resuspended in a reaction buffer with ATP to activate polymerase activity of Cas10. Polymerization products

(e.g., $cA_3$ and $cA_4$) are used for the downstream assays. **d** TtCsm$^{Csm3-D34A}$ polymerization reactions were performed with α-$^{32}$P-ATP as shown in (**c**) and products were resolved using thin-layer chromatography (TLC). Black arrow shows migration of solvent in the TLC plate. Bands were annotated using chemically synthesized standards (Supplementary Fig. 1d). 3 μL (−RNA capture) or 120 μL (+RNA capture) of SARS-CoV-2 N-gene RNA ($10^{10}$ copies/μL) diluted in total RNA of 293T cells were used for reactions. **e** TtCsm6-based fluorescent readout (top panel) is used for detection of $cA_4$ generated by TtCsm$^{Csm3-D34A}$ with (red bars) or without RNA capture step (blue bars) as shown in panel (**c**). SARS-CoV-2 N-gene RNA diluted in total RNA of 293T cells was used as a target. Fluorescence was measured with qPCR instrument and normalized to the no target control (NTC, 293T RNA only, dashed line). In each assay, the mean (n = 3) fluorescent signal was compared with one-way ANOVA. Pairwise comparisons with NTC were performed using post hoc Dunnett's test. Data are shown as mean ± SD. *$p < 0.05$; **$p < 0.01$; ***$p < 0.001$. Source data are provided as a Source Data file.

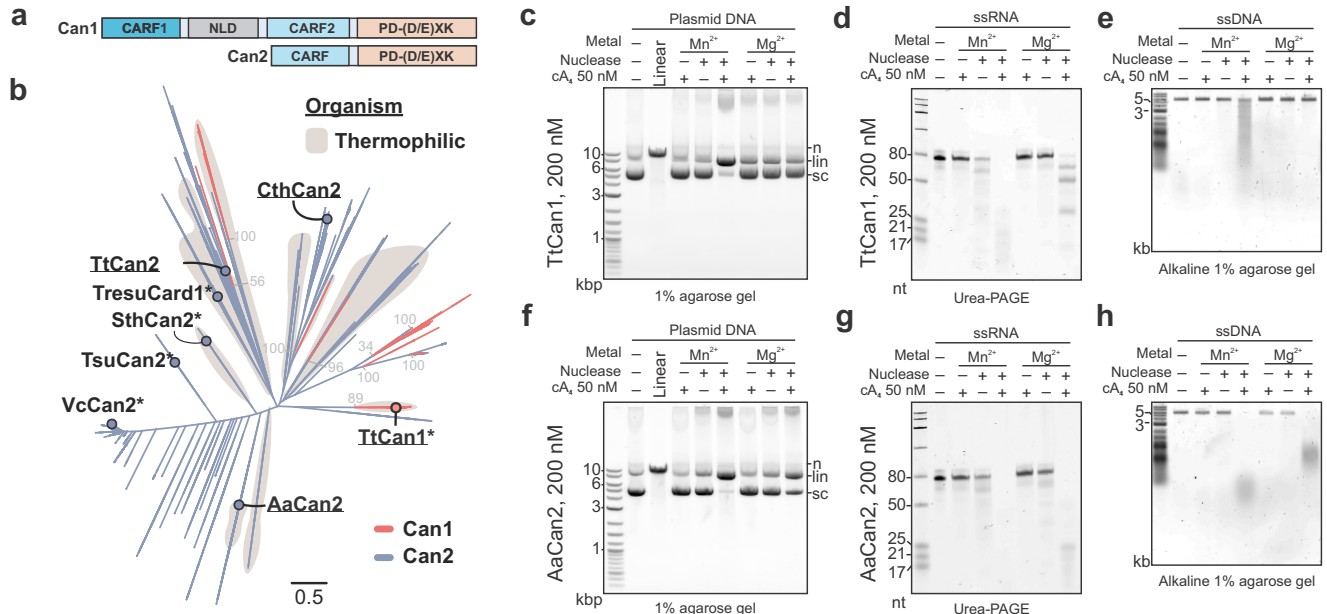

**Fig. 2 | Can1 and Can2 ancillary nucleases function as cA4-activated nucleases on plasmid DNA, ssRNA, and ssDNA. a** Domain organization of Can1 and Can2 proteins. Can2 proteins have two domains – CARF and PD-(D/E)XK superfamily nuclease domain. Can1 is predicted to be derived from Can2 by gene duplication[32]. NLD – nuclease-like domain. **b** Maximum-likelihood phylogeny of 204 Can1 (CARF2 and PD-(D/E)XK nuclease domain) and 3121 Can2 proteins. The light gray numbers indicate bootstrap values. Previously studied effectors are marked with asterisks (*). Effectors chosen for purification and in vitro experiments are underlined.

**c−e** Plasmid DNA (15 nM), ssRNA (425 nM), and ssDNA (15 nM) cleavage assay with TtCan1 (200 nM) in the presence of cA4 (50 nM). The reactions were incubated 15 min at 60 °C. n − nicked, lin − linear, sc − supercoiled plasmid. **f−h** Cleavage assays with AaCan2 (200 nM) in the presence of cA4 (50 nM). Assays were performed with 15 nM plasmid DNA, 425 nM ssRNA or 15 nM ssDNA for 15 min at 60 °C. n − nicked, lin − linear, sc − supercoiled plasmid. Source data are provided as a Source Data file.

type III CRISPR-based RNA capture and concentration results in the synthesis of more cyclic nucleotides, we added Csm-beads to 120 µL of a sample containing a mixture of SARS-CoV-2 RNA and total RNA extracted from HEK 293T cells (Fig. 1c, see "Methods"). After concentrating the beads with a magnet, we resuspended the pellet in a buffer containing α-$^{32}$P-ATP, allowed the cyclic polymerization to proceed, and analyzed the reactions using thin-layer chromatography (TLC). Type III CRISPR-based concentration increases the amount of cA3 and cA4, as compared to reaction performed without RNA concentration (Fig. 1c, d and Supplementary Fig. 1d).

Previously, we repurposed TtCsm6, a cA4-activated ribonuclease, to generate a real-time fluorescent readout for Csm-based RNA detection[18] (Fig. 1e, top). We reasoned that increased cA4 levels after RNA enrichment will boost the nuclease activity of TtCsm6 and therefore increase the sensitivity of the RNA detection. To test this hypothesis, we titrated $10^8$ to $10^5$ copies/µL of SARS-CoV-2 N-gene RNA into total RNA extracted from HEK 293T cells, concentrated the target RNA using TtCsm$^{Csm3-D34A}$, resuspended the beads in a buffer containing ATP, and then transferred the polymerization products to a reaction containing TtCsm6 and a fluorescent RNA reporter (i.e., FAM-RNA-Iowa Black FQ). Csm-based RNA enrichment increased the sensitivity of the assay 100-fold compared to the assay without the pull-down (Fig. 1e). Taken together, these results demonstrate how type III-A CRISPR-complexes can be used to capture sequence-specify RNAs, resulting in a higher concentration of cyclic nucleotides, which improves the sensitivity of sequence-specific RNA detection.

## CARF-nucleases Can1 and Can2 exhibit cA4-specific nuclease activities

Csm6 proteins contain an amino-terminal CARF (CRISPR-associated Rossman Fold) and a carboxy-terminal HEPN (Higher Eukaryotes and Prokaryotes Nucleotide-binding) domain[12,14]. Csm6 family proteins form homodimers, and the two CARF-domains bind cA4[25,29] or cA6[24],

which activate the C-terminal HEPN nuclease domain. However, the CARF domain of some Csm6 proteins also degrades the cyclic nucleotide, which inactivates the nuclease and may limit the sensitivity of Csm6-based assays[30]. To improve the sensitivity, we sought to identify and incorporate a CARF-nuclease that is activated by, but does not degrade, cA4.

CRISPR ancillary nucleases (Can) are another family of recently identified proteins that are activated by cyclic oligoadenylates and lack ring nuclease activity[31–33]. Like Csm6 proteins, Can proteins also contain amino-terminal CARF domains, but the carboxy-terminal nucleases are distinct. The Can1 protein from *Thermus thermophilus* (TtCan1) has a unique monomeric architecture with two non-identical CARF domains, one nuclease-like domain (NLD), and one restriction endonuclease domain (PD-(D/E)XK)[33], while Can2 nucleases contain a single CARF domain and form symmetrical homodimers[31,32] (Fig. 2a).

To identify Can1 and Can2 orthologs, we generated hidden Markov models (HMMs) to query publicly available microbial genomes and metagenomes from NCBI and JGI. This analysis identified 204 Can1 and 3121 Can2 proteins. Based on this analysis, we selected one Can1 protein and three Can2 proteins from thermophilic organisms. We hypothesized that thermostable nucleases will be compatible with RNA detection and target RNA-activated polymerization of ATP by the Csm complex from *Thermus thermophilus* (i.e., TtCsm). Elevated temperatures are anticipated to improve accessibility to RNA targets that might otherwise be obscured by secondary structures and the use of thermostable ancillary nucleases may enable detection of a specific RNA in a single tube at a single temperature. In addition, the stability of these proteins[34], may have downstream benefits when it comes to packaging and distribution.

Consistent with the previous research[33], TtCan1 exhibits Mn$^{2+}$-dependent linearization of supercoiled plasmid DNA in the presence of cA4 (Fig. 2c). Given the unique asymmetric interaction between the two non-identical CARF-domains of TtCan1 and cA4, we hypothesized that

other cyclic oligonucleotides without two-fold symmetry (e.g., cA₃ or cAAG) might allosterically activate the nuclease activity. We tested a library of twelve chemically synthesized cyclic oligonucleotides in a reaction with TtCan1 and three different substrates (i.e., dsDNA, ssRNA, and ssDNA) (Supplementary Fig. 2). Although none of the other cyclic oligonucleotides activate the TtCan1 nuclease, cA₄-activated TtCan1 robustly cleaves ssRNA and long but not short ssDNA substrates (Fig. 2d, e and Supplementary Fig. 2c). This pattern suggests that the nuclease activity of TtCan1 is either sequence-specific and the preferred cut site is not present in the short oligonucleotide or this nuclease requires a longer substrate for recognition and/or cleavage.

Can2 genes from *Clostridium thermobutyricum* (CthCan2), *Thermus thermophilus* (TtCan2), and *Archaeoglobi* archaeon JdFR-42 (AaCan2) were cloned and expressed in *E. coli* (Fig. 2b). However, only AaCan2 purified in quantities sufficient for biochemical assays (Supplementary Fig. 3). We systematically tested the activities of AaCan2 on different substrates with a range of cyclic oligoadenylates (Supplementary Fig. 4). Like TtCan1, AaCan2 also linearizes supercoiled plasmid DNA, and cleaves both ssRNA and ssDNA when activated with cA₄ (Fig. 2f–h and Supplementary Fig. 4). Cleavage of a ssDNA short oligos (71 nt) produces a discrete band suggesting that the enzyme processes ssDNA to a minimal cleavage product or that the activity is sequence-specific (Supplementary Fig. 4c). Collectively, our results demonstrate that TtCan1 and AaCan2 function as cA₄-activated nucleases on plasmid DNA, ssRNA, and ssDNA.

Both TtCan1 and AaCan2 linearize supercoiled DNA when activated with cA₄, which is followed by slow DNA degradation (Supplementary Fig. 5a). To determine if the initial cleavage event is sequence-specific, we sequenced plasmid DNA linearized with TtCan1 or AaCan2. To identify the cleavage site(s), we mapped the start and end positions of each read to the reference DNA sequence. This analysis identified that while both nucleases preferentially linearize the supercoiled DNA in two regions (1700–1750 and 5800–5850 bp), the most frequent TtCan1 cut sites map to poly-G sequences, while AaCan2 cuts at poly-A and poly-T sites, which indicates different cleavage preference of the two nucleases (Supplementary Fig. 5b, c).

While TtCan1 and AaCan2 rapidly digest supercoiled DNA with distinct sequence preferences, we do not detect cleavage at the same sites if DNA is linearized beforehand (Supplementary Fig 5d, e). Further, both nucleases digest single-stranded form of ΦX174 bacteriophage DNA to small fragments but cut at several locations in the supercoiled double-stranded replicative form (RF1) that has identical nucleotide sequence (Fig. 2e, h and Supplementary Fig. 5f). Taken together, this data suggests that TtCan1 and AaCan2 nucleases specifically act on single-stranded nucleic acids, but not on relaxed dsDNA. Collectively, this suggests that plasmid linearization is driven by cleavage in the underwound regions of the supercoiled DNA.

## Can2 ancillary nuclease provides sensitive Csm-based RNA detection
To determine if incorporating TtCan1 or AaCan2 improves sensitivity of the Csm-based RNA detection assay, we screened a library of short synthetic RNA and DNA reporters designed to identify sequences that might be preferred by these nucleases (see "Methods", Supplementary Data 1, and Supplementary Fig. 6). Consistent with our gel-based assays, cA₄-activated AaCan2 cleaves DNA reporters in the presence of Mn²⁺ and RNA reporters in the presence of either Mg²⁺ or Mn²⁺, but reactions with RNA reporter and Mn²⁺ consistently result in higher fluorescent signal (Supplementary Fig. 6a–c). While TtCan1 cleaves the same RNA reporters as AaCan2, cleavage by TtCan1 requires higher concentrations of cA₄ and produces less fluorescent signal (Supplementary Fig. 6d).

Having established that AaCan2 is more active than TtCan1, we set out to compare AaCan2 to the sensitivity of TtCsm6, which we used previously[18]. This comparison was performed by measuring cA₄ concentration-dependent activity for AaCan2 and TtCsm6 using the preferred RNA reporter and conditions that support the highest activity among tested variables (Supplementary Figs. 6, 7). AaCan2 produces a similar fluorescent signal to TtCsm6 when activated with 20-fold less cA₄ (0.5 nM versus 10 nM) (Fig. 3a, b). Moreover, AaCan2 exhibits an incremental decrease in cleavage rates with decreasing cA₄, while TtCsm6 exhibits a dramatic (non-linear) drop in the activity. This distinction in activity between the enzymes is consistent with the ring-nuclease activity of TtCsm6 rapidly degrading its activator, while AaCan2 binds and preserves the cyclic nucleotide (Supplementary Fig. 8).

Incorporating AaCan2 into the type III-based detection assay produces high background that is only evident in the presence of the TtCsm-complex, whereas AaCan2 alone demonstrates very little non-specific cleavage (Supplementary Fig. 9a and Fig. 3b). This disparity suggests that non-sequence specific activation of the Cas10 polymerase may generate low levels of cA₄, which stably activates AaCan2, whereas the ring-nuclease of TtCsm6 rapidly degrades cA₄ limiting the background signal. To test this hypothesis, we titrated concentration of TtCsm^Csm3-D34A-complex and show that 50-fold less of the complex (i.e., 25–0.5 nM) reduces the background without compromising target-specific activity, which significantly improves signal-to-noise ($p < 0.001$, Supplementary Fig. 9b, c).

Finally, we benchmarked AaCan2 and TtCsm^Csm3-D34A-complex combination against TtCsm6-based detection. The TtCsm6-based assay reliably detects $10^6$ copies/µL of target RNA (Fig. 3c), while AaCan2-based reactions are more sensitive ($10^5$ copies/µL) (Fig. 3d). Collectively, these results demonstrate that coupling AaCan2 with TtCsm^Csm3-D34A provides more sensitive RNA detection.

## Incorporating cA₃-dependent nuclease activity does not provide additional sensitivity of RNA detection
While our assay uses cA₄-activated collateral cleavage of ssRNA reporters, the TtCsm^Csm3-D34A complex also produces cA₃ (Fig. 1d and Supplementary Fig. 1d). We hypothesized that combining cA₃- and cA₄-sensing nucleases might enhance the sensitivity of TtCsm-based detection (Fig. 4a). NucC (Nuclease, CD-NTase associated) is an endonuclease activated by cA₃[26,35,36]. We purified three thermophilic NucC orthologs and tested cA₃-dependent dsDNA cleavage (Supplementary Fig. 10). The NucC from *Clostridium tepidum* (CtNucC) has the highest dsDNase activity and digests plasmid DNA into 300–400 bp fragments in the presence of cA₃ (Fig. 4b, left; Supplementary Fig. 11a). Deep sequencing of cleavage fragments determined that all purified NucC nucleases have a preference for 5′-ANNT-3′ sequence motif, which is consistent with previously published work[36] (Fig. 4b, right; Supplementary Fig. 11b–e).

Next, we set out to determine if CtNucC and AaCan2 could be combined into a single reaction to improve the sensitivity of RNA detection with TtCsm^Csm3-D34A. To perform fluorescent assays with CtNucC, we designed a 31 bp dsDNA reporter comprising six repeats of the optimal cleavage site (Supplementary Data 1). The lowest concentration of cA₃ detected by CtNucC is 0.05 nM (Fig. 4d and Supplementary Figs. 12, 13). However, TtCsm^Csm3-D34A coupled with CtNucC and dsDNA reporter only detects high concentrations of target RNA (i.e., $10^7$ copies/µL; Fig. 4e). Further, combining CtNucC with AaCan2 and matching fluorescent probes (i.e., dsDNA and ssRNA, respectively) (Fig. 4a) into a single reaction does not improve the sensitivity compared to detection with AaCan2 alone (Fig. 4f, g and supplementary Fig. 13). While CtNucC is sensitive to cA₃ activation, the TtCsm-complex may not produce sufficient concentrations of this cyclic nucleotide to increase sensitivity over AaCan2 detection alone.

## Type III CRISPR-based RNA capture and detection directly from clinical samples
Most CRISPR-based diagnostics assays reported to date use purified RNA as the starting material for viral detection assays[37]. The cost of

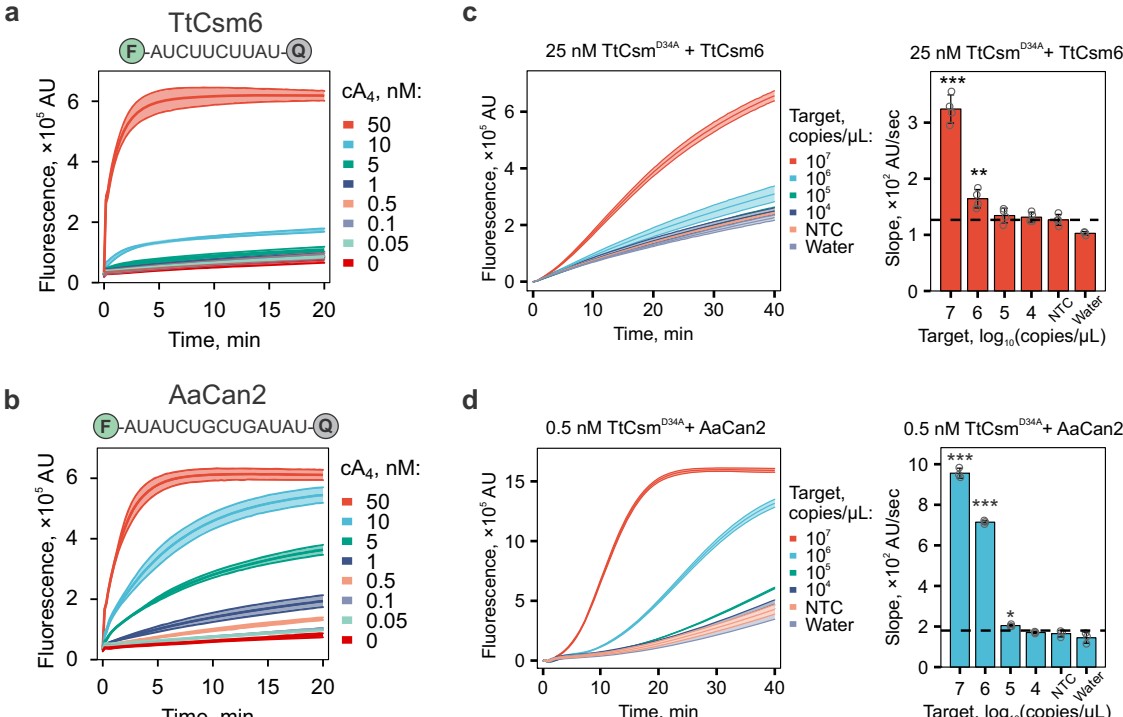

**Fig. 3 | Incorporation of Can2 nuclease improves sensitivity of Csm-based RNA detection. a** TtCsm6 (300 nM) and **b** AaCan2 (300 nM) cleavage assays with fluorescent ssRNA reporter (top) in the presence of varying $cA_4$ activator concentrations (shown with colors). Data are shown as the mean (center line) of three replicates ± S.D. (ribbon). The optimal fluorescent reporter (top) was determined using RNA library screen in Supplementary Fig. 6. TtCsm RNA detection coupled with TtCsm6- (**c**) and AaCan2-based (**d**) assays were performed using samples with target RNA concentrations ranging from $10^7$ to $10^4$ copies/μL. Samples were prepared by spiking IVT fragments of SARS-CoV-2 N gene into total RNA extracted from nasopharyngeal swab patient sample negative for SARS-CoV-2. Cleavage of fluorescent RNA reporter was detected by measuring fluorescence every 10 s in a real-time PCR instrument (left). Data were plotted as mean of four replicates. Simple linear regression was used to calculate slopes for linear regions of the curves. Bars show mean values ($n = 4$) ± S.D. (right). Data was analyzed with one-way ANOVA followed by multiple comparisons to NTC sample using one-tailed post-hoc Dunnett's test. ***$p < 0.001$; **$p < 0.01$; *$p < 0.05$. AU = arbitrary units. Source data are provided as a Source Data file.

RNA extraction, time to extract the RNA, and specialized equipment necessary to extract the RNA is often not discussed. We hypothesized that thermostable type III CRISPR complexes (e.g., TtCsm) could be used to capture target RNA directly from patient samples, without prior RNA extraction. Sequence-specific capture and concentration directly from patient samples could drastically reduce time, reagents, and equipment costs.

To address this challenge, we set out to incorporate "capture and concentration" into the fluorescent-based detection method. However, the magnetic beads used to pull-down target RNA unexpectedly obscure the fluorescent signal released when AaCan2 cleaves the reporter (Supplementary Fig. 14d). Initially, we sidestepped the problem by performing a second magnetic pull-down after the polymerization and transferring the products (without the beads) to a cleavage reaction with the ancillary nuclease (Fig. 1e and Supplementary Fig. 14a, b). However, this introduces an additional liquid handling step (three steps total), complicating the assay. In addition, reactions with AaCan2 following RNA pull-down produce high background (Supplementary Fig. 14b), which results in a low signal-to-noise ratio. We hypothesized that both limitations could be resolved by reducing the concentration of the beads used for RNA capture. To test this hypothesis, we titrated Csm-beads in the pull-down and then combined the polymerization reaction and AaCan2 nuclease into a single step (Supplementary Fig. 14c, d). Reducing the concentration of Csm-beads 100-fold rescues fluorescent signal, improves signal-to-noise, and streamlines the assay by eliminating the need for additional sample handling.

Next, we tested six lysis buffers supplemented with detergents (i.e., Triton X-100 or NP-40), and chelating agents (i.e., EDTA or EGTA), to identify lysis conditions compatible with the two-step detection assay that relies on Csm-based RNA capture, followed by detection with AaCan2. The lysis buffers tested do not inhibit the assays with purified RNA and increase the signal-to-noise ratio (Supplementary Fig. 15). This improvement may be a consequence of the detergents, which according to the manufacturer, reduce aggregation of the magnetic beads. Further, when we titrate the viral RNA, the optimized two-step assay generates a fluorescent signal that is significantly different from the negative control (293T RNA) at target concentrations ≥8.3 fM ($5 \times 10^3$ copies/μL, Fig. 5a–c).

To determine the sensitivity of direct SARS-CoV-2 RNA detection in patient samples, we used two unprocessed SARS-CoV-2 positive patient samples that were serially diluted in a negative patient swab sample (Fig. 5d). Each sample was divided in two parts. One part (120 μL) was tested directly with the two-step type III RNA detection assay, and the other part (125 μL) was used for RNA extraction. Purified RNA was tested with the two-step type III detection protocol (120 μL) and RT-qPCR (5 μL) (Fig. 5d). Csm-based RNA capture assay detects SARS-CoV-2 RNA in both unprocessed patient swabs and purified RNAs with Ct <24, which corresponds to ~8.7 × 10³ copies/μL (~15 fM) (Fig. 5e).

Finally, to test cross-reactivity of TtCsm-based RNA capture coupled with AaCan2-based fluorescent detection, we use a panel of seven respiratory viruses, including SARS-CoV-1, Middle East respiratory syndrome coronavirus (MERS-CoV), seasonal coronaviruses 229E (HCoV-229E), NL63 (HCoV-NL63), and HKU1 (HCoV-HKU1). To perform the assay, we used the highest concentration of genomic RNAs

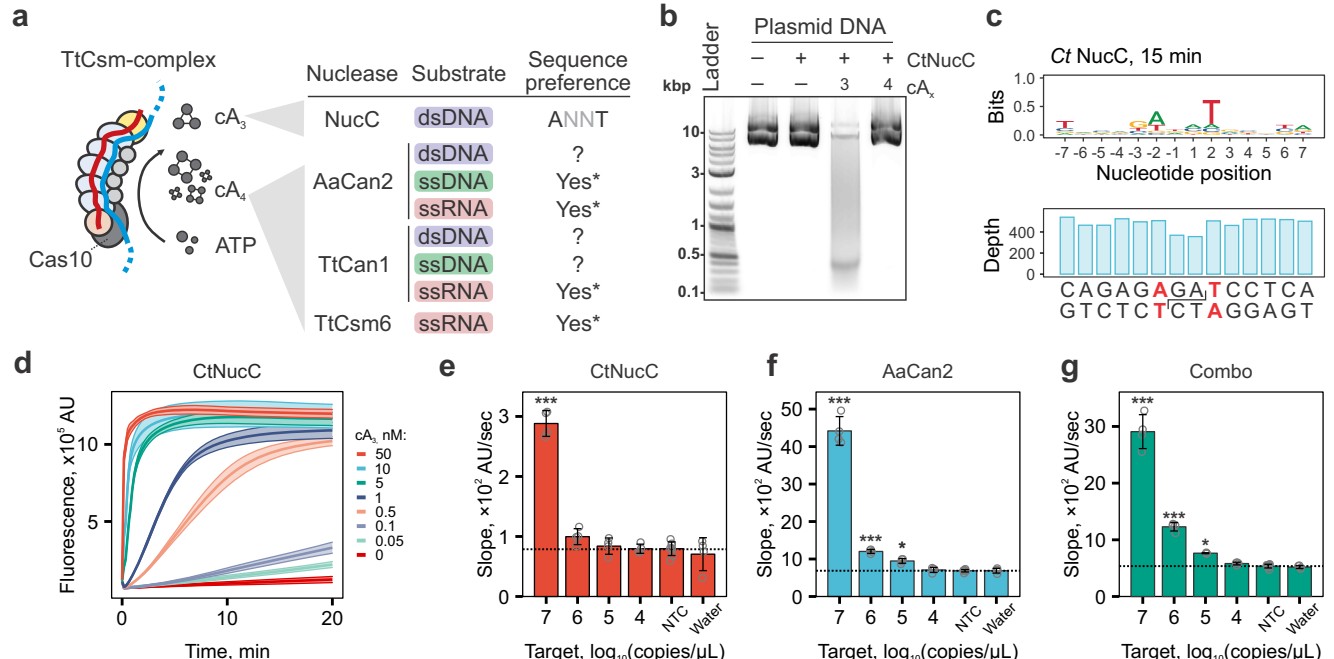

**Fig. 4 | Incorporation of cA₃-activated nucleases into Csm-based RNA detection assay. a** The target bound TtCsm complex primarily generates cA₄ and cA₃. Schematic summarizes cA₄- and cA₃-dependent activities of nucleases biochemically tested. N/D – not detected; Asterisks (*) indicate nucleases that have sequences preferences (Supplementary Fig. 6). **b** CtNucC (15 nM) is activated by cA₃ (20 nM) and cleaves plasmid DNA into short fragments in 15 min. **c** Deep sequencing of DNA fragments generated after 5 min of incubation with CtNucC revealed the preferential cleavage sites (ANNT). The reduced sequencing depth at the cut site is consistent with a cleavage mechanism producing 3′-overhangs that are removed by T4 DNA polymerase when sequencing library is prepared. **d** CtNucC (300 nM) cleavage assay with fluorescent dsDNA reporter across eight concentrations of cA₃ (shown with colors). Data is shown as mean (center line) of four replicates ± S.D. (ribbon). **e**–**g** TtCsm RNA detection assays coupled with CtNucC

(dsDNA reporter that include preferred cleavage sequence), AaCan2 (ssRNA reporter), and combination of AaCan2 and CtNucC (both reporters). Reactions were performed using samples with target RNA concentrations ranging from $10^7$ to $10^4$ copies/μL. Samples were prepared by spiking IVT fragment of SARS-CoV-2 N gene in total RNA of SARS-CoV-2 negative nasal swab. Cleavage of the fluorescent reporter was detected by measuring fluorescence every 10 s in a real-time PCR instrument. Simple linear regression was used to determine slopes for four replicates. See Supplementary Fig. 13 for fluorescent curves used in the analysis. Data were plotted as mean ($n = 4$) ± S.D. and analyzed with one-way ANOVA. All samples were compared to the non-target RNA control (NTC) using one-tailed post-hoc Dunnett's test. ***$p < 0.001$; **$p < 0.01$; *$p < 0.05$. AU – arbitrary units. Source data are provided as a Source Data file.

available at ATCC and recommended by FDA (~5 × 10⁵ copies/μL) (https://www.fda.gov/medical-devices/coronavirus-disease-2019-covid-19-emergency-use-authorizations-medical-devices/in-vitro-diagnostics-euas). No cross-reactivity is detectable with RNAs from Influenza B, Human respiratory syncytial virus (RSV), and HCoV-229E (Fig. 5f). However, we found that reactions with SARS-CoV-1, MERS-CoV, HCoV-HKU1, and HCoV-NL63 RNAs generate a weak signal that is slightly higher than the threshold in negative control samples. Importantly, this signal is significantly lower than the signal obtained for SARS-CoV-2 RNA used at the same concentration ($p < 0.001$, one-way ANOVA with posthoc Tukey HSD test).

## Discussion

Development of CRISPR-based diagnostics has primarily focused on type V (Cas12) and type VI (Cas13) CRISPR systems, and the sensitivity of these techniques has improved from picomolar[38] to attomolar concentrations[30] over the last six years. However, most CRISPR-based viral diagnostics described to date still require nucleic acid extraction and pre-amplification to reach clinically relevant sensitivities[4].

While sensitivity continues to improve, less attention and little progress has been made to develop methods that bypass RNA extraction. Purifying nucleic acids from patient samples prior to testing requires specialized equipment that increases the cost, labor, and time-to-result. Thus, testing protocols that require RNA extraction represent a major limitation for point-of-care implementation, which is critical for limiting the spread of a contagious agent. Here, we demonstrate how type III CRISPR systems can be used for sequence-

specific capture and detection of viral RNA directly from unprocessed patient samples without pre-amplification.

In 2021, the first attempts to repurpose type III CRISPR systems for SARS-CoV-2 diagnostics achieved 0.1–1 nM sensitivity of RNA detection without pre-amplification[18,19]. More recent improvements using different type III complexes or different ancillary nucleases have been used to detect spiked SARS-CoV-2 RNA with ~2–4 fM sensitivity in 30 min[20,26] (Supplementary Fig. 17). However, the reported time-to-result for most CRISPR diagnostics does not account for the time (i.e., 20–40 min) or cost (i.e., $5–7) of extracting RNA, which varies depending on the technique, availability of reagents, accessibility of high complexity laboratories, reliability of shipping, and the number of samples. Here, we contribute to the ongoing development of type III-based diagnostics by developing a method for sequence-specific capture and concentration of target RNAs directly from unprocessed patient samples. This approach enables direct detection of $8.7 × 10^3$ copies of SARS-CoV-2 RNA per μL (~15 fM) in clinical samples without laboratory-based RNA extraction or pre-amplification in less than 20 min total. While the sensitivity of the approach is still inferior to RT-qPCR, it is sufficient to identify infected individuals capable of spreading SARS-CoV-2[39] and is comparable to rapid antigen tests[2].

Like Cas13, type III systems also recognize RNA, and the most sensitive detection methods developed to date for either approach rely on collateral nuclease activity to release a fluorescent signal[4]. While methods that incorporate tandem CRISPR nucleases (i.e., Cas13 and Csm6) are currently more sensitive (~50 aM), the intrinsic amplification of RNA recognition by type III systems may ultimately improve

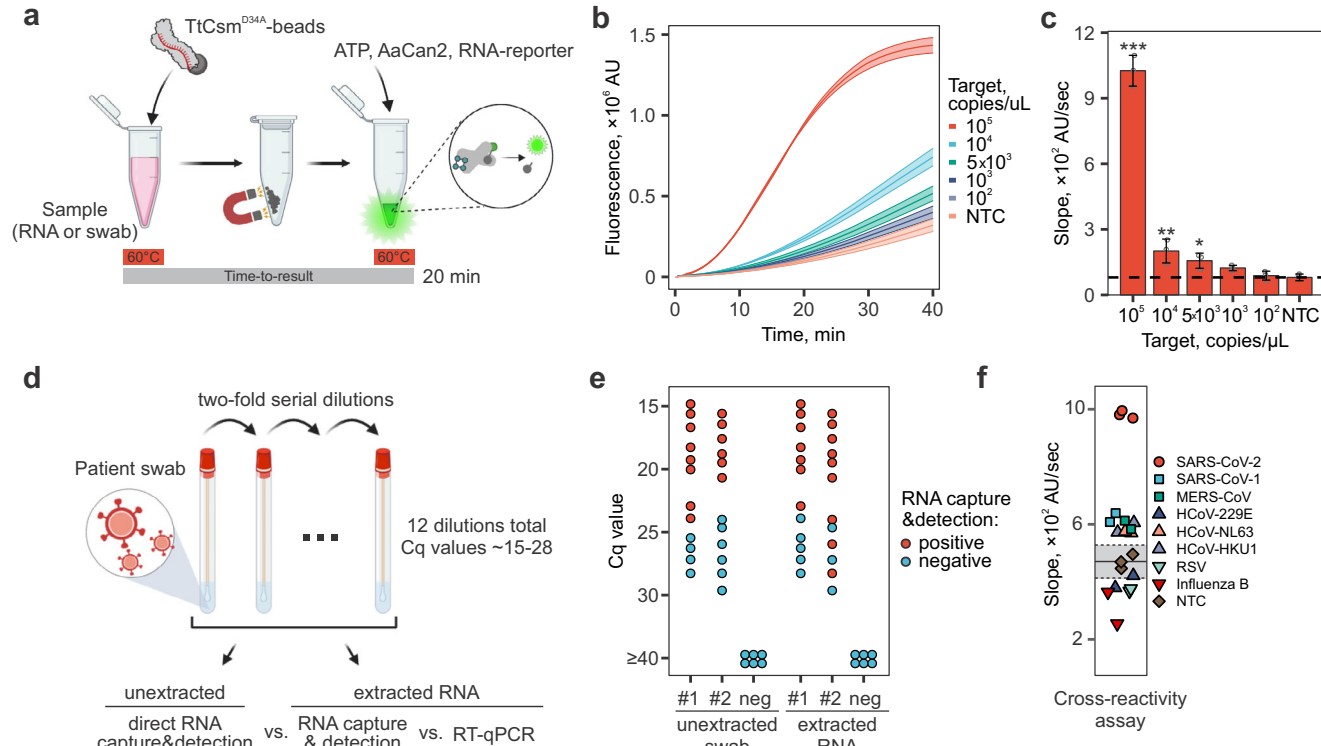

**Fig. 5 | TtCsm-based RNA capture directly detects SARS-CoV-2 in clinical samples. a** Schematic of TtCsm-based RNA capture assay from patient sample coupled with AaCan2-based fluorescent detection. Two-step protocol includes incubation of TtCsm-beads with RNA sample at 60 °C followed by concentration of the magnetic beads, and addition of a buffer containing ATP, AaCan2 and a fluorescent RNA reporter (i.e., FAM-RNA-Iowa Black FQ). **b** RNA detection assay as shown in (**a**) was performed using samples with target RNA concentrations ranging from $10^7$ to $10^2$ copies/μL. Samples were prepared by spiking IVT fragments of SARS-CoV-2 N gene into in total human RNA (HEK 293T cells). Cleavage of fluorescent RNA reporter was detected by measuring fluorescence every 10 s in a real-time PCR instrument (left). Data were plotted as mean of three replicates ± S.D. (ribbon). **c** Simple linear regression was used to calculate slopes for linear regions of the curves. Bars show mean values ($n = 3$) ± S.D. (right). Data was analyzed with one-way ANOVA followed by multiple comparisons to NTC sample using one-tailed post-hoc Dunnett's test. ***$p < 0.001$; **$p < 0.01$; *$p < 0.05$. **d** Patient samples (e.g., nasopharyngeal swabs)

tested positive for SARS-CoV-2 RNA with RT-qPCR were serially diluted in a negative patient swab sample and dilutions were tested with three different methods. Non-processed swabs were tested directly with TtCsm-based RNA capture & detection assay, while purified RNAs were tested with RT-qPCR and TtCsm-based RNA capture & detection assay. **e** Twenty-four nasopharyngeal mock samples were prepared from two patient swabs (#1, Ct-14.8 and #2, Ct-15.6) and tested as shown in (**d**). Red dots show samples positive in type III detection, blue dots show negative samples. **f** RNAs of common respiratory viruses and coronaviruses related to SARS-CoV-2 ($5 \times 10^5$ copies/μL) were tested with TtCsm-AaCan2 RNA capture and detection assay (three-step protocol). Reaction kinetics were measured using a real-time PCR instrument, and slopes were quantified in the linear range of the fluorescence curves. Dotted line shows mean of three negative samples (NTC). Mean ± 2.33 S.D. for negative samples is shown with gray rectangle. AU – arbitrary units. Source data are provided as a Source Data file.

sensitivity. Type III systems uniquely amplify RNA recognition in two sequential steps: first, through Cas10-mediated polymerization of cOAs, and second, through cOA-mediated activation of multi-turnover effectors (e.g., Can). In addition to the advantages that might come from consecutive stages of signal amplification, the separation of target recognition by the type III surveillance complex (i.e., Csm or Cmr) from collateral cleavage by ancillary effectors also enables programmable RNA capture. Unlike Cas13, which relies on the same active site for target and non-target collateral cleavage[40], the RNase-dead TtCsm complex (TtCsm$^{Csm3-D34A}$) can capture and maintain target RNA from a larger volume and concentrate these RNAs for various downstream applications. Incorporating RNA capture increases the sensitivity of type III CRISPR-based diagnostic and allows direct detection in clinical samples without RNA extraction, a prerequisite for most current platforms. Further, the separation of on-target (cis-) and collateral (trans-) activities in type III CRISPR systems provides a flexible platform for creating combinatorial assemblies of detection (i.e., Csm/Cmr complexes) and readout (i.e., ancillary effectors) modules based on diverse CARF-, SAVED-(SMODS-associated and fused to various effector domains)[16] or CBASS (Cyclic oligonucleotide-based antiphage signaling system) proteins for applications in diagnostics and beyond.

Direct detection of RNA in clinical samples without RNA extraction or pre-amplification will advance direct-to-consumer diagnostics. However, eliminating pre-amplification removes additional selectivity imparted by the traditional use of primers. To evaluate cross-reactivity of TtCsm-based RNA capture coupled with AaCan2-based fluorescent detection, we tested a panel of seven respiratory viruses. At high RNA concentrations ($5 \times 10^5$ copies/μL) we detect a weak, but reproducible signal to some of the coronaviruses (i.e., SARS-CoV-1, MERS-CoV, HCoV-HKU1, and HCoV-NL63) but not to HCoV-229E or other respiratory viruses (i.e., RSV, Influenza B). Each of the coronaviruses have 4–7 mismatches in first two segments (S1–S2) of crRNA:target duplex (Supplementary Fig. 18), which permit target RNA binding but significantly decreases polymerization activity of the Cas10 subunit[19]. Coronaviruses that generate a weak signal have two mismatches in the first segment (S1; nucleotides +2 and +5), while HCoV-229E, which generates no signal, has an additional mismatch at the 1st position. The first two nucleotides in segment S1 have the greatest effect on cOA production[19,26,41], therefore the additional mismatch in HCoV-229E at the 1st position might explain the complete loss of signal, as compared to the weak signal generated by the other coronaviruses, which contain two mismatches in segment 1 (S1). Our original bioinformatic pipeline for crRNA design filtered out guides with potential cross-reactivity,

however, we prioritized the total number of mismatches in segments S1 and S2 rather than the position of these mismatches[18]. The cross-reactivity assay in this manuscript and published biochemical data[19,26,41] demonstrate the importance of mismatch positions +1 and +2, which is expected to improve the specificity of the next generation of guides. Overall, we anticipate that type III-based RNA pull-down techniques that bypass RNA extraction, combined with further optimization of lysis conditions, more efficient guide design, and the integration of next generation of signal detection methods (e.g., real-time sequencing, digital enzymology, amperometry, etc.) will help to bring type III CRISPR diagnostics closer to current standards of rapid molecular testing.

## Methods

### Ethical statement

The study was reviewed by the Montana State University Institutional Review Board (IRB) For the Protection of Human Subjects (FWA 00000165). De-identified clinical samples were obtained with IRB approval (protocol #DB033020) and informed consent from patients undergoing testing for SARS-CoV-2 at Bozeman Health Deaconess Hospital.

### Human clinical sample collection and preparation

Nasopharyngeal swabs from patients that either tested negative or positive for SARS-CoV-2 were collected in viral transport media. RNA was extracted from all patient samples using the QIAamp Viral RNA Mini Kit (QIAGEN).

### Nucleic acids

Sodium salts of cyclic di-, tri-, tetra-, penta- and hexa-adenosine monophosphates ($cA_{2-6}$), sodium salts of cyclic tri- and tetra- guanosine monophosphates ($cG_2$, $cG_3$), sodium salts of cyclic (guanosine- (2′ –> 5′)- monophosphate- adenosine- (3′ –> 5′)- monophosphate) (cGAMP), cyclic adenosine monophosphate-guanosine monophosphate (cAG), and cyclic diguanosine-5′-monophosphate ($cG_2$) were purchased from Biolog Life Science Institute. Fluorescent reporters (RNA and DNA) were purchased from IDT (Supplementary Data 1). The dsDNA reporter was ordered as a duplex from IDT. Target and non-target RNAs of SARS-CoV-2 N-gene were in vitro transcribed with MEGAscript T7 (Thermo Fisher Scientific) from PCR products generated from pairs of synthesized overlapping DNA oligos (Supplementary Data 1) (Eurofins). Transcribed RNAs were purified by denaturing PAGE. Total RNA from HEK 293T cells was extracted using TRIzol reagent. The genomic RNAs of SARS-CoV-1, Middle East respiratory syndrome coronavirus (MERS-CoV), seasonal coronaviruses 229E (HCoV-229E), NL63 (HCoV-NL63), and HKU1 (HCoV-HKU1) were purchased from ATCC.

### Non-targeting control (NTC)

Total RNA extracted from SARS-CoV-2 negative nasopharyngeal swabs or total RNA extracted from HEK 293T cells (ATCC, CRL-3216) were used as negative controls. RNA extracted from HEK 293T cells was diluted to match the average Ct level (~27) obtained for RNAseP mRNA in RNA samples extracted from nasopharyngeal swabs (Supplementary Table 1). The RT-qPCR for RNase P mRNA was performed using CDC RP primers and probe (2019-nCoV CDC EUA Kit, IDT#10006606).

### Plasmids

Plasmids encoding the type III-A Csm complex from *Thermus thermophilus* (pCDF-5xT7-TtCsm; Addgene #128572 and pACYC-TtCas6-4xcrRNA4.5; Addgene #127764), were a gift from Jennifer Doudna. Vector pCDF-5xT7-TtCsm was used as a template for site-directed mutagenesis to mutate the D34 residue in Csm3 to alanine (D34A) and inactivate Csm3-mediated cleavage of target RNA (pCDF-5xT7-TtCsmCsm3-D34A)[42]. The CRISPR array in pACYC-TtCas6-4xcrRNA4.5

was replaced with a synthetic CRISPR array (GeneArt) containing five repeats and four identical spacers, designed to target the N-gene of SARS-CoV-2 (i.e., pACYC-TtCas6-4xgCoV2N1)[18]. TtCas6 was PCR was PCR-amplified from the pACYC-TtCas6-4xcrRNA4.5 plasmid and cloned between the NcoI and XhoI sites in the pRSF-1b backbone (Millipore Sigma) (pRSF-TtCas6). Expression vector encoding TtCsm6 nuclease, pC0075 TtCsm6 His6-TwinStrep-SUMO-BsaI, was a gift from Feng Zhang (Addgene plasmid #115270)[43].

Gene fragments encoding for Can1 from *Thermus thermophilus* (TtCan1; "WP_011229147.1"), Can2 from *Archaeoglobi archaeon* JdFR-42 (AaCan2; "2730024700"), *Clostridium thermobutyricum* (CtCan2; "WP_195972101.1"), and *Thermus thermophilus* (TtCan2; "WP_143585921.1"), were codon optimized for expression in *E. coli*, synthesized by GenScript, and cloned into pC0075 vector (Addgene #115270) in frame with the N-terminal His6-TwinStrep-SUMO tag using NcoI and XhoI restriction sites to replace the TtCsm6 gene. NucC from *Clostridium tepidum* BSD2780120874b_170522_A10 (CtNucC; "WP_195923598.1"), *Elioraea* sp. Yellowstone (EsNucC; "WP_141855040.1") and *Acidimicrobiales bacterium* mtb01 (Amtb01NucC; "TEX45487.1"), were cloned into pC0075 backbone using the same restriction sites as for Can1 and Can2 genes.

### Protein expression and purification

Expression and purification of the TtCsm^Csm3-D34A complex and TtCsm6 were performed as previously described[18]. TtCan1, AaCan2, CtCan2, TtCan2, CtNucC, EsNucC, and Amtb01NucC) were purified according to the following protocol. Each expression vector was transformed into *Escherichia coli* BL21(DE3) cells and grown in LB Broth (Lennox) (Thermo Fisher Scientific) at 37 °C to an OD600 of 0.5. Cultures were then incubated on ice for 1 h, and then induced with 0.5 mM IPTG for overnight expression at 16 °C. Cells were lysed with sonication in Lysis buffer (20 mM Tris-HCl pH 8, 500 mM NaCl, 1 mM TCEP) and lysate was clarified by centrifugation at $10,000 \times g$ for 25 min, 4 °C. The lysate was heat-treated at 55 °C for 45 min and clarified by centrifugation at $10,000 \times g$ for 25 mins at 4 °C. His$_6$-TwinStrep-tagged protein was bound to a StrepTrap HP column (Cytiva) and washed with Lysis buffer. The protein was eluted with Lysis buffer supplemented with 2.5 mM desthiobiotin and concentrated (10k MWCO Corning Spin-X concentrators) at 4 °C. Affinity tags were removed from the protein using His-tagged SUMO protease (100 μL of 2.5 mg/mL protease per 20 mg of protein) during dialysis against SUMO digest buffer (30 mM Tris-HCl pH 8, 500 mM NaCl, 1 mM dithiothreitol (DTT), 0.15% Igepal) at 4 °C overnight. The tag and the protease were applied to HisTrap HP column (Cytiva), and the flow-through was concentrated using Corning Spin-X concentrators at 4 °C. Finally, the protein was purified using a HiLoad Superdex 200 26/600 size-exclusion column (Cytiva) in storage buffer (20 mM Tris-HCl pH 7.5, 1 mM DTT, 400 mM monopotassium glutamate, 5% glycerol). Fractions containing the target protein were pooled, concentrated, aliquoted, flash-frozen in liquid nitrogen, and stored at −80 °C.

### $^{32}$P-labeling of RNA oligos

Target (SARS-CoV-2 N1) and non-target RNAs were transcribed from PCR extended duplex oligos using home-made T7 RNA polymerase (Supplementary Table 2) (Eurofins). The IVT RNAs were gel purified and dephosphorylated with Quick CIP (NEB) for 20 min at 37 °C in 1× CutSmart Buffer (NEB). The phosphatase was inactivated by heating at 80 °C for 5 min before 5′ end-labeling the RNAs with T4 polynucleotide kinase (NEB) and [γ-$^{32}$P]-ATP (PerkinElmer) for 30 min at 37 °C. The kinase was heat inactivated by heating at 65 °C for 20 min.

### Binding and pull-down of RNA oligos with TtCsm

For the experiments shown in Fig. 1b and Supplementary Fig. 1b, c, $^{32}$P-labeled RNA (25 nM) was incubated with TtCsm^Csm3-D34A (160 nM) targeting SARS-CoV-2 N-gene in 1× Binding Buffer (25 mM HEPES, pH

7.5, 150 mM NaCl, 1 mM TCEP) for 20 min at 65 °C. The reaction mixtures were added to 10 μL of HisPur Ni-NTA Magnetic beads (ThermoFisher) equilibrated in Binding Buffer and incubated on ice 30 min with vortexing every 10 min. The beads were separated from the supernatant using a magnet and washed with 50 μL 1× binding buffer. The RNA was extracted from supernatant (unbound fraction) and beads (bound fraction) using Acid Phenol: chloroform (Ambion). Extracted RNA was resolved using UREA-PAGE, exposed to a phosphor screen, and imaged on a Typhoon 5 imager (Amersham). Bands corresponding to the IVT RNAs were quantified using ImageJ v1.52t and the percent bound calculated [bound/(bound + free)*100%].

### Complexing of TtCsm with magnetic beads
The HisPur Ni-NTA Magnetic beads (ThermoFisher) were washed two times with a 1× Binding Buffer (25 mM HEPES, pH 7.5, 150 mM NaCl, 1 mM TCEP). For one reaction, 5 μL of equilibrated beads were mixed with TtCsm$^{D34A}$ complex (25 nM) in 1× Binding Buffer (V = 50 μL) and incubated for 30 min on ice. The beads with the complex (Csm-beads) were concentrated with a magnet and resuspended in 5 μL of 1× Binding Buffer.

### Thin-layer chromatography (TLC)
For the experiments shown in Fig. 1c, 3 μL of positive sample (target RNA diluted in NTC, $10^{10}$ copies/μL) or 3 μL of NTC were mixed with TtCsm$^{Csm3-D34A}$ complex (25 nM) and 250 μM ATP supplemented with [α-$^{32}$P]-ATP (PerkinElmer) in the reaction buffer (20 mM Tris-HCl pH 7.8, 250 mM monopotassium glutamate, 10 mM ammonium sulfate, 1 mM TCEP (tris(2-carboxyethyl)phosphine)), 5 mM magnesium sulfate). The reaction was incubated at 60 °C for 1 h. For the pull-down reactions, 120 μL of positive or negative samples were mixed with 5 μL of Csm-beads in Binding Buffer (25 mM HEPES, pH 7.5, 150 mM NaCl, 1 mM TCEP) for 10 min at 60 °C. The Csm-beads were concentrated with a magnet and the supernatant was discarded. The Csm pellets were resuspended in 30 μL of the reaction buffer and 250 μM ATP supplemented with [α-32P]-ATP (PerkinElmer). Reaction products were phenol-chloroform extracted and resolved on silica TLC plates (Millipore).

Samples (1 μL) were mixed with 100 mM sodium acetate, pH 5.2 (2 μL), and spotted 1.5 cm above the bottom of the TLC plate. The plate was placed inside a 2 L beaker filled to -0.5 cm with developing solvent (0.2 M ammonium bicarbonate pH 9.3, 70% ethanol, and 30% water) and capped with aluminum foil. The plate was run for 2 h at room temperature and dried. TLC plate was exposed to a phosphor screen and imaged with Typhoon phosphor imager. Chemically synthesized standards (2 μM) were resolved on the same TLC plate and visualized using UV shadowing.

To test cA$_3$ and cA$_4$ hydrolysis in the presence of ancillary nuclease, radiolabeled cA$_3$ and cA$_4$ produced above were mixed with nuclease (500 nM) in the reaction buffer and incubated for 1 hour at 55 °C. Reaction products were phenol-chloroform extracted and resolved using TLC for 45 min as described above.

### Type III-based RNA detection
3 μL of RNA sample was mixed with 250 μM ATP, 0.5 nM TtCsm$^{D34A}$ complex, 300 nM of nuclease (TtCsm6, AaCan2, or CtNucC) with corresponding reporter in a reaction buffer (20 mM Tris-HCl pH 7.8, 150 mM or 250 mM monopotassium glutamate, 10 mM ammonium sulfate, 1 mM TCEP (tris(2-carboxyethyl)phosphine)), 5 mM magnesium sulfate (for TtCsm6 and CtNucC) or 5 mM manganese(II) chloride (for AaCan2) in a 30 μL reaction. The reporter B8 (300 nM) was used for the reaction with TtCsm6, D7 (300 nM) – with AaCan2, and dsDNA probe (300 nM) – with CtNucC. Reactions were incubated at 55 °C or 60 °C. Cleavage of fluorescent reporters was detected by measuring fluorescence every 10 sec in a real-time PCR instrument QuantStudio 3 (Applied Biosystems).

### Type III-based RNA pull-down and detection
To bind TtCsm$^{D34A}$ complex with the magnetic beads, the HisPur Ni-NTA Magnetic beads (ThermoFisher) were washed two times with a 1× Binding Buffer (25 mM HEPES, pH 7.5, 150 mM NaCl, 1 mM TCEP). 5 μL of equilibrated beads were mixed with TtCsm$^{D34A}$ complex (30 nM) in 1× Binding Buffer (V = 50 μL) and incubated for 30 min on ice. The beads with the complex (Csm-beads) were concentrated with a magnet and resuspended in 5–500 μL of 1× Binding Buffer. For one reaction 5 μL of resuspended Csm-beads was used.

**Pull-down and detection from RNA samples and nasopharyngeal swabs.** 120 μL of sample was mixed with 5 μL of Csm-beads in 1× Binding Buffer for 10 min at 60 °C (in experiments shown in Fig. 5 of 1× Binding Buffer was supplemented with 0.01% Triton X-1001 mM EDTA). For pull-downs from nasopharyngeal swabs RNase Inhibitor (N2615, Promega) was added to the final concentration 40 U/μl. Csm-beads were concentrated with a magnet and the supernatant was discarded. For three-step protocol the Csm-beads pellet was resuspended in 20 μL of the 1× reaction buffer (20 mM Tris-HCl pH 7.8, 250 mM monopotassium glutamate, 10 mM ammonium sulfate, 1 mM TCEP (tris(2-carboxyethyl)phosphine)), 5 mM magnesium sulfate / manganese(II) chloride) containing ATP (250 μM). The reaction was incubated at least 10 min at 60 °C, the Csm-beads were pelleted, and the supernatant (10 μL) was transferred to a new reaction with TtCsm6 (300 nM) and B8 RNA Reporter (300 nM) or AaCan2 (300 nM) and D7 RNA Reporter (300 nM) in 1× reaction buffer (V = 30 μL) (Supplementary Data 1). Reactions were incubated at 55 °C. For two-step protocol the Csm-beads pellet was resuspended in 30 μL of the 1× reaction buffer (20 mM Tris-HCl pH 7.8, 250 mM monopotassium glutamate, 10 mM ammonium sulfate, 1 mM TCEP (tris(2-carboxyethyl)phosphine)), 5 mM magnesium sulfate/manganese(II) chloride) containing ATP (250 μM), RNA Reporter (300 nM), and nuclease AaCan2 (300 nM). The reaction was incubated at least 10 min at 60 °C. Cleavage of the fluorescent RNA reporter was detected by measuring fluorescence every 10 s in a real-time PCR instrument QuantStudio 3.

### RT-qPCR
RT-qPCR was performed using N1 and RP CDC primers (2019-nCoV CDC EUA Kit, IDT#10006606). RNA was extracted from patient samples with QIAamp Viral RNA Mini Kit (QIAGEN, # 52906) and used for one-step RT-qPCR in ABI 7500 Fast Real-Time PCR System according to CDC protocols (https://www.fda.gov/media/134922/download). In brief, 20 μL reaction included 8.5 μL of Nuclease-free Water, 1.5 μL of Primer and Probe mix (IDT, 10006713), 5 μL of TaqPath 1-Step RT-qPCR Master Mix (ThermoFisher, A15299) and 5 μL of the RNA. Nuclease-free water was used as negative template control (NTC). Amplification was performed as follows: 25 °C for 2 min, 50 °C for 15 min, 95 °C for 2 min followed by 45 cycles of 95 °C for 3 s and 55 °C for 30 s. To quantify viral RNA in the samples, standard curve for N1 primers was generated using a dilution series of a SARS-CoV-2 synthetic RNA fragment (RTGM 10169, NIST) spanning N gene with concentrations ranging from 10 to $10^6$ copies per μL. Three technical replicates were performed at each dilution. The NTC showed no amplification throughout the 45 cycles of qPCR.

### Nanopore sequencing of DNA cleavage fragments
DNA cleavage fragments were sequenced using Oxford Nanopore with Ligation Sequencing Kit (SQK-LSK109). After incubation with nucleases, cleavage fragments were column-purified using DNA Clean & Concentrator-5 kit (Zymo Research, D4004) as instructed. Next, for each sample 200 ng of purified DNA was used to prepare sequencing libraries with NEBNext® Ultra™ II DNA Library Prep Kit (NEB, E7645S). Briefly, DNA was end-repaired with NEBNext Ultra II End Prep Enzyme Mix, which fills 5′- and removes 3′- overhangs. Next, end-repaired fragments were barcoded with Native Barcoding Expansion kit (ONT, EXP-NBD104) using Ultra II Ligation Master Mix (NEB). Barcoded DNA

fragments were pooled together and purified with magnetic beads (Omega Bio-tek, M1378-01). Freshly mixed 80% ethanol was used to wash magnetic bead pellet. Sequencing adapters (AMII) were ligated to barcoded DNA using NEBNext® Quick Ligation Module (NEB, E6056S). Ligation reactions were purified with magnetic beads. SFB buffer (ONT, EXP-SFB001) was used for washes. Resulting DNA library was eluted from the beads in 20 μL of EB buffer (QIAGEN, #19086). DNA concentration was measured with Qubit dsDNA HS Assay (ThermoFisher, Q32851), and 20 ng was loaded on the Nanopore MinION (MIN-101B, R9.4.1 flow cell). The flow cell was primed, and library was loaded according to Oxford Nanopore protocol (SQK-LSK109 kit). The sequencing run was performed in the fast base calling mode in the MinKNOW software v5.2.4.

## Sequencing data analysis

Sequenced reads were demultiplexed using guppy-barcoder v6.2.11+e17754edc (ONT) and aligned with minimap2 v2.17-r954-dirty (-ax map-ont mode) to the reference plasmid sequence that was modified by adding 1000 bp overlaps at the 5′- and 3′- ends. Overlapping regions were introduced to account for circular nature of the plasmid. Resulting alignments (BAM files) were sorted and indexed using samtools v1.13. Next, *bamtobed* function in bedtools package v2.30.0 was used to generate BED files and read coordinates were extracted. Read end coordinates were used to calculate cleavage fragment length distributions and map frequencies of cuts at specific locations. To analyze the sequence preferences of each nuclease, 14 bp windows surrounding read ends were extracted with *getfasta* function from bedtools package v2.30.0. Resulting fasta files were used to calculate position weigh matrices (PWMs) with *getPwmFromFastaFile()* function in DiffLogo v2.16.0R package. Finally, PWMs were plotted as sequence logos using ggseqlogo v0.1R package. Sequencing depth around the most frequent cut site for each nuclease was calculated with samtools v1.13 *depth* function and plotted with ggplot2 package v3.3.5 in RStudio v2022.07.1+554.

## RNA and DNA reporter's libraries

To determine the optimal RNA or DNA reporter for each cOA-activated nuclease, we constructed a library of variable single-stranded RNA or DNA molecules tethering a FAM fluorophore to an Iowa Black quencher. The Biostrings package in R was used to construct a library of reporter sequences containing each of the 64 unique trinucleotide combinations possible. Since multiple unique trinucleotides could be included in a single reporter (e.g., 5′-FAM-AUAGAAGAAU-IABkFQ-3′ contains AGA, GAA, and AAG), we narrowed our initial library of 64 reporters to remove redundant sequences. This resulted in a library of 24 unique reporter sequences, each of which were integrated into reporters of different length (Supplementary Data 1). The R-script used to design these reporters is accessible on GitHub (WiedenheftLab/RNA_reporter_design, https://doi.org/10.5281/zenodo.7368892).

## In vitro DNA and RNA cleavage assays

All reactions were performed in a buffer containing 20 mM Tris-HCl pH 7.8, 50–250 mM monopotassium glutamate, 10 mM ammonium sulfate, 1 mM TCEP, 5 mM magnesium sulfate or 5 mM manganese chloride. Plasmid DNA cleavage assays were performed by incubating 15 nM of Lenti-luciferase-P2A-Neo (Addgene #105621) plasmid with TtCan1, AaCan2 or CtNucC (15–200 nM) in the presence of cOAx (20–50 nM) in 10 μL reaction. After 5–15 min incubation, Gel Loading Dye, Purple (6X) (NEB) was added and 4 μL was loaded on 1% agarose gel. For ssDNA and ssRNA cleavage assays, 0.425 μM of 71 nt DNA oligo (CGTCGTACCGGTTAGAGGATGGTGCAAGCGTAATCTGGAACATCGT ATGGGTATGCCCACGGTGTCCACGGCG, Eurofins), 0.425 μM of 74 nt IVT RNA SARS-CoV-2 N-gene (Supplementary Table 2) or 15 nM of ssDNA ΦX174 bacteriophage (NEB, N3023) were incubated with TtCan1 (200 nM) or AaCan2 (200 nM) in the presence of cOAx

(20–50 nM) in 10 μL. After 5–15 min incubation, 10 μL was loaded on 12% UREA PAGE or 1% alkaline agarose gel.

## Phylogenetic analysis of Can1 and Can2 proteins

A DELTA-BLAST was initiated, using previously described Can1 and Can2 proteins as queries[31–33] to generate individual lists of closely related proteins with an e-value cutoff of $10^{-4}$ and 50% query coverage. The resulting sequences were then used as queries to initiate a PSI-BLAST search with an E-value cutoff of $10^{-4}$ and 50% query coverage. This step was repeated until convergence and redundant sequences were removed with CD-HIT v4.7[44]. In case of Can1, sequences from a previously published dataset[16] that contain two CARF domains and a nuclease domain were used to generate multiple sequence alignment of Can1-related proteins. In total, 29 sequences of Can1-related proteins and 2,531 sequences of Can2-related proteins were used separately to generate multiple sequence alignment with a local version of MAFFT v7.429[45] (−localpair −maxiterate 1000). The generated alignments for Can1 and Can2 were curated with MaxAlign v1.1[46] to remove misaligned or non-homologous sequences. The resulting dataset—comprised of 29 Can1-like and 1,283 Can2-like proteins, respectively—were then individually realigned with MAFFT and HMMbuild[47] (HMMER v3.2.1) was used to generate HMM profiles from each alignment. The resulting profiles were used to search a local database of prokaryotic genomes from NCBI (downloaded on June 11, 2021) and list of sequences identified in BLAST search from previous steps. An initial search performed with these HMM profiles identified 1442 Can1 and 5,431 Can2 homologs, which were manually filtered according to the presence of domains that define each protein, as well as the presence of conserved residues found in CARF and nuclease domains. The resulting set of 204 Can1 and 3,121 Can2 proteins were merged into a single file and aligned in MAFFT (LINSI option) for downstream phylogenetic analyses. Next, Trimal v1.4[48] was used to remove columns in the alignment comprised of ≥70% gaps. Thermostable homologs of Can1 and Can2 were annotated according to organisms that they are originated. ProtTest v3.4.2[49] was used to select an evolutionary model, and a phylogenetic tree was constructed in IQ-TREE v1.6.1[50] using the recommended model (i.e., LG+G+F). The phylogenetic tree was plotted using the ggTree package in R[51].

## Phylogenetic analysis of NucC

A phylogenetic tree of NucC proteins was generated using the same methods as described above for Can1/Can2 proteins. Briefly, DELTA-BLAST and PSI-BLAST searches with previously identified NucC proteins[35] generated a list of closely related proteins (e-value cutoff of $10^{-4}$ and minimum 50% query coverage). The resulting dataset was filtered with CD-HIT v4.7 to remove redundant sequences. The resulting 1230 NucC sequences were aligned with MAFFT (−localpair −maxiterate 1000), and poorly aligned and highly gapped sequences were removed with MaxAlign. The resulting set of 896 NucC sequences were re-aligned with MAFFT as previously described, and the resulting alignment was used to generate a NucC HMM profile which we used to search within prokaryotic genomes from NCBI. This search identified 1774 hits, which were filtered according to the presence of restriction endonuclease-like domain (i.e., $ID_{x30}EAK$-motif containing), gate-loop and $cA_3$ binding domains and were aligned with MAFFT. The remaining NucC homologs were curated according to organisms they are originated from to identify thermostable NucC homologs. The resulting alignment of 1510 NucC proteins with 21 thermostable homologs was used to generate a phylogenetic tree with FastTree v2.1.10[52] and was plotted using the ggTree package in R.

## Statistics & reproducibility

All statistical analyses were performed in RStudio. Analysis of Variance Models (ANOVA) were calculated with *aov()* function in the stats R package. Multiple comparisons between positive samples and negative controls were performed using Dunnett's test with multcomp R

package. Reaction slopes were determined by extracting coefficients from linear models fitted to fluorescence data with *lm()* function in R. The linear regions of the fluorescence curves were identified using rolling regression with *auto_rate()* function in respR package. Statistical threshold for detecting SARS-CoV-2 in patient samples with Csm-based assay was set as mean of negative control ± 2.33S.D., which captures 98% of variation in negative samples (2% false positive). Statistical significance levels used in the figures are \*\*\*$p < 0.001$, \*\*$p < 0.01$, and \*$p < 0.05$. No statistical method was used to predetermine the sample size. The experiments were not randomized, and the investigators were not blinded to allocation during experiments and outcome assessment. No data were excluded from the analyses.

### Reporting summary

Further information on research design is available in the Nature Portfolio Reporting Summary linked to this article.

### Data availability

Raw reads for Nanopore sequencing are available at Sequence Read Archive under BioProject accession number "PRJNA907862". Phylogenetic analysis data generated in the current study are available on Wiedenheft lab GitHub page (https://github.com/WiedenheftLab/; https://doi.org/10.5281/zenodo.7368902, https://doi.org/10.5281/zenodo.7369225). Source data are provided with this paper.

### Code availability

Custom codes are available on GitHub (https://github.com/WiedenheftLab/) and Zenodo repositories: designing RNA and DNA reporter libraries (https://doi.org/10.5281/zenodo.7368892); phylogenetic analyses (https://doi.org/10.5281/zenodo.7368902, https://doi.org/10.5281/zenodo.7369225); Nanopore sequencing data analysis (https://doi.org/10.5281/zenodo.7374621).

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

## Acknowledgements

We thank members of Bozeman Health who provided deidentified patient samples. This work was supported by National Institutes of Health (United States) grants 1K99AI171893-01 (A. Nemudryi) and 1K99GM147842 (A.S-F.). A.S.-F. is a postdoctoral fellow of the Life Science Research Foundation that is supported by the Simons Foundation. A.S.-F. is supported by the Postdoctoral Enrichment Program Award from the Burroughs Wellcome Fund. Research in the Wiedenheft lab is supported by the NIH (R35GM134867), the M.J. Murdock Charitable Trust, a young investigator award from Amgen, a generous gift from the Rosolowsky family, and the Montana State University Agricultural Experimental Station (USDA NIFA). The Koutmou lab's contributions to this work were supported by the NIH (R35GM128836). Funders had no role in designing, performing, interpreting, or submitting the work. Figures were created with BioRender.com.

## Author contributions

B.W., A. Nemudraia, A. Nemudryi, and A.S.-F. conceived the experimental plans. B.W., A. Nemudraia, and A. Nemudryi supervised the research. A. Nemudraia, A. Nemudryi, and R.W. developed Type III Csm-based RNA concentration method. A.M.S., T.Z., R.W., M.B., J.N., S.P., A. Nemudraia and A.S.-F. purified the proteins and performed biochemical characterization. A. Nemudryi performed statistical analyses and analyzed sequencing data. L.R., J.J., and K.K. contributed to the initial design of TLC assays. L.H. and A. Nemudryi performed TLC; M.B., S.P., and T.W. performed the bioinformatic analyses and phylogenetics. H.L. and A.M. performed RNA extractions and RT-qPCR of patient nasopharyngeal swab samples. A. Nemudraia and A. Nemudryi performed RT-qPCR and Csm-based detection assay. A.M.S. and M.B. contributed equally to the manuscript. A. Nemudraia, A. Nemudryi, and B.W. wrote the manuscript. All authors edited and approved the manuscript.

## Competing interests

B.W. is the founder of SurGene LLC, and VIRIS Detection Systems Inc. B.W., A. Nemudryi, A. Nemudraia, and A.S.-F. are inventors of the patent applications US17/240858 and PCT/US2021/029219 pertaining to use of engineered type III CRISPR-Cas system for sensitive and sequence-specific detection of nucleic acids filed by Montana State University. The remaining authors declare no competing interests.
