## [Peer Review File · Nature Communications]

REVIEWER COMMENTS

Reviewer #1 (Remarks to the Author):

Nemudraia et al continues the authors' pioneering work on adapting the thermophilic type III CRISPR-Cas system for SARS-CoV-2 RNA detection applications. Two main improvements are made in this new work, particularly to improve detection sensitivity such that the CRISPR reactions can be performed with prior nucleic acid amplification: 1) the affinity-tagged RNase-dead Csm complex is used as a handle to concentrate target RNAs prior to triggering cyclic adenylate (cA) generation; and 2) Csm-mediated cA generation is coupled to a choice of thermophilic nuclease effectors to further amplify signal.

I really enjoy reading the biochemical characterization parts of the manuscript. The dual dsDNA/ssRNA cleavage activity of Can enzymes and their substrate specificity switch dictated by signaling cA molecules are intriguing, and the effort to combine cA3/4-sensing nucleases to amplify Csm-based signals is creative. However, I think further biochemical insight is needed for the RNA detection application to be substantially improved. The current detection assay on SARS-CoV-2 RNA just did not perform well. The authors claimed to push the detection sensitivity to femtomolar ranges, but the signal-to-noise of detection at this conc. range is poor (Fig. 2g). The femtomolar sensitivity range (and clinical sensitivity at Ct~21) is also too poor for robust SARS-CoV-2 RNA detection. I also have questions about the detection specificity which is not convincingly demonstrated in the manuscript.

- It is unclear why Can and Nuc orthologs were specifically chosen from thermophilic organisms. Is this so that the Can/Nuc-mediated reaction can be combined in the same tube and incubate at high temperatures along with the TtCsm reaction (this seems to be implied at Line 155)? Are there other benefits to performing these reactions at higher temperatures, such as less interference from RNA secondary structures? The authors should state these motivations more clearly
- Related to the point above, have the authors tried combining the cA production and fluorescent detection in the same tube? Even if this does not work well it would be good to know, for future improvements.
- It was unclear how the authors arrived at specific reaction conditions for the newly discovered thermophilic orthologs of Can and Nuc, and whether there were optimizations needed. The authors should explicitly show and discuss these efforts.
- Related to the point above about the reaction conditions, a sentence like Line 204-205 (comparing performance of TtCan1 and AaCan2) is not very meaningful, unless we know that both enzymes are operating at their optimal conditions. Is 60 deg optimal for Can1, and 55 deg for Can2? Are there differences in oligomeric states of Can1 and Can2 which binds to cAs at different molar ratios?
- Fig 3d: is it possible that NucC can degrade cA4 (though it was previously shown in Gruschow et al to not degrade cA3)? This might explain why supplementing NucC to the Csm/Can2 reaction has a negative effect.
- Re: the lysis buffer composition assay, it is better to demonstrate this with an AaCan2-based readout instead of the TtCsm6-based readout currently shown. The more sensitive AaCan2 might be inhibited by different detergents or conditions compared to those affecting Csm6.
- The authors performed a cross-reactivity assay in their previous publication (Santiago-Frangos et al 2021, which used LAMP in combination with TtCsm) but did not do so here. This is an experiment that should be repeated, as it is not certain the high specificity of detection would be maintained in this LAMP-free, TtCsm/AaCan2-based detection.
- Line 20 ("we make two major advances that simultaneously limit sample handling and significantly enhance the sensitivity...") is too much of an oversell. The TtCsm/AaCan2

assay still involves multiple liquid handling steps, a bead enrichment step, supernatant removal, resuspension, liquid transfer, etc.

- The high background observed in negative controls is worrisome, and I do not think resorting to reporting slopes is robust enough for clinical diagnosis, especially as multiple liquid/bead handling steps (which can affect individual component concentrations in the reaction, and therefore affect kinetics/slopes) are involved. If the background source is non-sequence specific activation of Cas10 to generate low-level cA4 as the authors suggested (Line 226), is there a way to overcome this?

- Related to the point above, the authors might need to do further biochemical investigations on factors that may affect the tool's sensitivity and specificity. Is Cas10 capable of producing enough cA to saturation during the reaction incubation, and what are the rate differences of cA generation from sequence-specific vs non-sequence specific activation? Do the nucleases consume all cAs generated by Cas10?

Minor point

- Supplementary Fig 2h has not be referred to in the manuscript (it should be mentioned at around Line 174)

Reviewer #2 (Remarks to the Author):

The emergence of the COVID-19 pandemics prompted a fast development of novel diagnostic methods. Class 2 CRISPR-Cas effector complexes as exemplified by Cas12 and Cas13 have been recently harnessed for the molecular diagnostics and development of the point-of-care tests for virus detection. Several attempts were also made to repurpose type III CRISPR systems for the diagnostics of SARS-CoV-2. In 2021, Wiedenheft's lab showed that the Type III effector complex of *Thermus thermophilus* (TtCsm) coupled with the cyclic-oligoadenylate (cA) dependent Csm6 nuclease could be harnessed for the viral RNA detection however detection sensitivity needed to be improved. In this manuscript, Nemudraia et al. sought to increase the sensitivity of TtCsm-based SARS-CoV-2 RNA detection to enable virus detection directly in the patient swab. The authors used two different approaches to improve sensitivity. First, they showed that His-tagged RNase-dead TtCsm bound to viral RNA could be concentrated using nickel-derivatized magnetic beads that results in the increased sensitivity and enables SARS-CoV-2 RNA detection in the nasopharyngeal swab without RNA extraction. However, further optimization was still required to improve sensitivity. Therefore, next Nemudraia et al. probed different cyclic-oligoadenylate dependent nucleases, Can1 and Can2 (cA4-dependent), and NucC (cA3-dependent), for the downstream signal amplification and systematically tested their activation using different cOAs. Unexpectedly, they found that Can1 nuclease from *T. thermophilus* and the Can2 from *Archaeoglobi* archaeon (Aa) are activated both by cA3 and cA4, and that different cAs trigger different substrate specificity (dsDNA or ssRNA, respectively). Since TtCsm complex produces both cA3 and cA4, authors hypothesized that combining cA3- and cA4- sensing nucleases might enhance the sensitivity of TtCsm-based detection. However, coupling of RNase-dead TtCsm complex with two downstream nucleases CtNucC and AaCan2 did not improve the sensitivity compared to the individual nucleases.

Major points:

1) Authors propose that various cyclic oligoadenylates, e.g., cA3 and cA4 produced by the TtCsm trigger different nucleic acid specificities providing a fine-tuned mechanism to regulate ancillary nuclease activity in response to the phage infection. However, their conclusion is based on a single point reaction. To support their hypothesis authors have to perform a detailed cleavage analysis of dsDNA, ssDNA or ssRNA to demonstrate clearly

which substrate is most favorable for Can1/Can2 nucleases.

2) Authors show that Can1 nuclease from *T. thermophilus* and the Can2 from *Archaeoglobi* archaeon are activated both by cA3 and cA4. However, structural studies of Can1, Can2 (and Can2-related nuclease Card1) unambiguously show that binding site for cOA in CARF domain is arranged to accommodate cA4 (but not cA3). Authors should explain how TtCan1 and AaCan2 become activated by an asymmetrical cA3 molecule to degrade dsDNA? What are binding affinities of cA4 and cA3 in this case?

3) Authors claim that reproducible cleavage of ssDNA was detected for AaCan2, together with the ssRNase and dsDNase activities (Fig. 2d, Supplementary Fig. 3d). However, data presented in Figure 2d and Supplementary Figure 3d are confusing: ssDNA is cleaved by AaCan2 (Supplementary Figure 3d) but almost no cleavage is observed in gel presented in Figure 2d.

Minor comment:

1) Supplementary Figure 3e: please specify "TtCsm6" above the gel, similar to "AaCan2".

Reviewer #3 (Remarks to the Author):

SUMMARY

The authors describe a series of experiments to improve the sensitivity and broader applicability of Type III CRISPR diagnostics. Using TtCsm capture and concentrate, the authors improve the sensitivity of Type III diagnostics and enhance this further with the characterization and implementation of accessory proteins (Can1, Can2, NucC). The authors use a nucleic acid biochemistry to profile the Can1 and Can2 accessory nucleases associated with Type III CRISPR-Cas systems. Strikingly, the authors demonstrated that the nature of the secondary messenger cyclic oligo adenylate (C3 or C4) leads to differential nuclease activity. This would suggest that the ligand binding to the CARF domain can allosterically switch the nuclease domain into one of two differing catalytically competent states - an observation that will be of great interest to the field. The manuscript is a valuable contribution to the development of CRISPR diagnostics and contains insights into the biology and mechanism of accessory proteins. Overall, the paper is well written, and the results explained clearly within the text. However, the Figures are somewhat difficult to understand relative to the methods and these should be carefully reviewed to improve clarity before publication.

COMMENTS

The authors note an issue with diagnostics sensitivity (specific to Type III systems) and suggest a capture and concentrate strategy using catalytically deficient TtCsm. Concentrating the RNA using magnetic beads to improve the sensitivity of the diagnostic is an innovative approach but this raises some questions and concerns about specificity. Could the authors comment on how they think capture and concentrate might affect off-target RNA binding and activation of the system? Is the TtCsm complex so specific that it would reject non-target RNA binding? Or is it likely that at very low target concentrations (i.e., those that are clinically relevant) the TtCsm complex may also enrich for off-target RNA (bearing some degree of affinity for the complex) which may in turn raise the chances of an off-target activation of the system?

FIGURES GENERAL

The authors include a lot of gel images (which is not a problem) but they are cropped and placed side by side where the impression is that they are the same method/stain/label/gel and that one ladder from one gel panel is appropriate or sufficient to interpret the adjacent cropped image. For example, Fig. 2C shows plasmid DNA (agarose) and ssRNA/ssDNA (UREA-PAGE) cleavage gel panels. But it's not clear how these are visualized. Are the

nucleic acids tagged? Was the gel post stained? The authors are strongly encouraged to consider including schematics or appropriate text in the Figure or legend to help the reader understand what the experiment is showing.

Fig. 2B: The phylogenetic tree has no scale and needs one. Same for Supplementary Fig. 6A

Dear Reviewers,

Thank you for the critical review of our manuscript. Your collective feedback, helpful suggestions, and healthy skepticism led us to re-test the biochemical activities of Can1 and Can2 nucleases, which identified contamination in the original protein stocks. Clearly, this was disappointing, but it highlights the value of peer review, and we are fortunate that the questions presented during the review helped us identify the problem. We have added your anonymous contribution to the acknowledgments.

To avoid being repetitive in our responses, we provide a short summary of the most substantial changes to the manuscript, which now includes 5 main figures, 18 supplemental figures and 3 tables. This summary is followed by point-by-point responses to each reviewer individually.

1. cA₃-dependent activities are not reproducible. In the original submission, we reported that the substrate specificity of Can1 and Can2 switches in response to the cyclic oligoadenylate activator (i.e., cA₃ vs cA₄). These results were extremely robust, highly reproducible, and the biological/biotechnological implications were very exciting. During the revision, we purified new stocks of Can1 and Can2 proteins to perform the requested experiments. The cA₄-dependent activities of both Can1 and Can2 were consistent with activities reported in the original paper but attempts to repeat cA₃-dependent activation failed. As you might imagine, this triggered considerable concern, and we invested significant effort to figure out what went wrong. Initially, we presumed that these failures were due to a technical error, so we repeated the experiments and failed again. After several failures, we started again (from the beginning) using all new reagents, new plasmid preps, new transformations, etc. We were persistent, in part because the dual cA₃/cA₄ activation was exciting, the results seem to add new biological insight, and (frankly) we were/are concerned about our scientific reputation (*please see the response to **Reviewer#2** where we explain in detail why cA₃-dependent activation of TtCan1 seemed plausible*). After numerous "fresh starts" by several individuals, we can only speculate that the original DNA stocks used to make Can1 and Can2 proteins were contaminated with another plasmid (i.e., NucC). It's an unsatisfying answer, but we are unable to come up with anything that can explain why/how this exceptionally robust activity disappeared.

2. Can1 and Can2 exhibit cA₄-specific nuclease activities. In summary, additional experiments now show (from multiple independent preps) that both TtCan1 and AaCan2 are cA₄-dependent nucleases but not activated by cA₃ (revised Fig. 2). The identity of these enzymes has been confirmed by mass spectrometry.

3. Improved diagnostic assay. While cA₃-stimulated activities of AaCan2 have been removed, we hope that the revised text and additional advances warrant consideration. Type III CRISPR-based capture and concentration of specific RNA 1) increases the diagnostic sensitivity ~100 times, 2) eliminates the need for time-intensive and expensive RNA extraction methods, which is a current limitation for most diagnostics, and 3) can be applied to any type III-based diagnostics beyond the TtCsm-based assay used in our study. Further, in the revised manuscript, we improve the originally presented technology by:

- simplifying the procedure (2 steps instead of 3) (**Fig. 5** and **Supplementary Fig. 14**)
- decreasing the time-to-result by 40% (33 to 20 min, **Fig. 5a**)
- improving the signal-to-noise ratio (**Supplementary Fig. 9**).
- adding new cross-reactivity data (**Fig. 5f**)

Other major changes to the revised manuscript include:

- we repeated *all* cleavage and RNA detection assays with Can1 and Can2 (revised **Figs. 2-5; Supplementary Figs. 2, 4 and 5**)
- we repeated RNA detection assays that combine Can2 and NucC nucleases (revised **Fig 4**)
- we re-tested clinical samples with the improved diagnostic assay (**Fig. 5d-f**)
- we added optimization data for each of the tested nucleases (**Supplementary Figs. 7 & 12**)
- we performed deep sequencing to determine linearization sites in plasmid DNA cleaved with Can1 and Can2

We hope these practical and conceptual warrant publication in *Nature Communications*. Thanks again for your time, effort, and valuable feedback.

Point-by-point responses to Reviewers' comments:

Reviewer #1:

Nemudraia et al continues the authors' pioneering work on adapting the thermophilic type III CRISPR-Cas system for SARS-CoV-2 RNA detection applications. Two main improvements are made in this new work, particularly to improve detection sensitivity such that the CRISPR reactions can be performed with prior nucleic acid amplification: 1) the affinity-tagged RNase-dead Csm complex is used as a handle to concentrate target RNAs prior to triggering cyclic adenylate (cA) generation; and 2) Csm-mediated cA generation is coupled to a choice of thermophilic nuclease effectors to further amplify signal.

I really enjoy reading the biochemical characterization parts of the manuscript. The dual dsDNA/ssRNA cleavage activity of Can enzymes and their substrate specificity switch dictated by signaling cA molecules are intriguing, and the effort to combine cA3/4-sensing nucleases to amplify Csm-based signals is creative. However, I think further biochemical insight is needed for the RNA detection application to be substantially improved. The current detection assay on SARS-CoV-2 RNA just did not perform well. The authors claimed to push the detection sensitivity to femtomolar ranges, but the signal-to-noise of detection at this conc. range is poor (Fig. 2g). The femtomolar sensitivity range (and clinical sensitivity at Ct~21) is also too poor for robust SARS-CoV-2 RNA detection. I also have questions about the detection specificity which is not convincingly demonstrated in the manuscript.

Thank you for characterizing our work as "pioneering". We agree that type III CRISPR-diagnostics have a unique and important role to play, and we hope that this paper will help advance this nascent field. In the revised manuscript, we performed additional optimizations that significantly improved signal-to-noise and increased clinical sensitivity from Ct ~21 to Ct ~24 (for details, see pts. 2, 8 and 9) (revised **Fig 5**). We acknowledge that the current assay is only one more step toward a clinically deployable diagnostic, but we think the advances reported in this manuscript

are conceptually and practically important, even if there are opportunities to continue improving the LoD.

1. It is unclear why Can and Nuc orthologs were specifically chosen from thermophilic organisms. Is this so that the Can/Nuc-mediated reaction can be combined in the same tube and incubate at high temperatures along with the TtCsm reaction (this seems to be implied at Line 155)? Are there other benefits to performing these reactions at higher temperatures, such as less interference from RNA secondary structures? The authors should state these motivations more clearly.

As the referee points out, we intentionally selected nucleases from thermophilic organisms for compatibility with the thermophilic TtCsm-complex. This allows combining TtCsm-based ATP polymerization and nuclease reporter cleavage reactions in a single tube at a single temperature (**Fig. 3c, d, Fig. 4d, and Fig. 5**). The initial focus on thermophilic proteins was two-pronged. First, as the reviewer suggested, elevated temperatures are anticipated to improve accessibility to RNA targets that might otherwise be obscured by secondary structures. Second, thermophilic proteins generally have higher stability (10.1110/ps.062130306), which may have downstream benefits when it comes to packaging and distribution. We acknowledge that the latter benefits are way down the road and are not the focus of our current work.

We revised the corresponding section of the paper to better articulate our reasoning for choosing thermostable Can1/2 and NucC orthologs (Lines 153-163).

2. Related to the point above, have the authors tried combining the cA production and fluorescent detection in the same tube? Even if this does not work well it would be good to know, for future improvements.

Reactions in **Figs. 3 & 4** (no upstream RNA concentration step) combine target-RNA activated TtCsm polymerase activity production with fluorescent detection in the same tube at a single temperature.

However, the magnetic beads used to pull down TtCsm and the corresponding target RNA interferes with our fluorimeter and obscured the signal (revised **Supplementary Fig. 14d**). Initially, we sidestepped this problem by transferring the products (without the beads) to a cleavage reaction with the ancillary nuclease (**Fig. 1e** and revised **Supplementary Fig. 14a, b**). However, as the referee points out, this introduces an additional liquid handling step (three steps total), complicating the assay.

In the revised manuscript, we addressed this technical limitation by reducing the concentration of Csm-beads in the pull-down step. We found that lowering bead concentration 100-fold (~0.02 µg/µL in the detection reaction) eliminates interference with the fluorimeter, rescuing fluorescent detection, and eliminates the need for an additional liquid transfer (**Supplementary Fig. 14c, d**). The updated assay now includes RNA extraction and concentration (step 1) and one-pot fluorescent detection (step 2) that combines cA production with fluorescent detection in the same tube at a single temperature (**Fig. 5a**). Importantly, lowering the concentration of TtCsm doesn't have a negative impact on the LOD.

3. It was unclear how the authors arrived at specific reaction conditions for the newly discovered thermophilic orthologs of Can and Nuc, and whether there were optimizations needed. The authors should explicitly show and discuss these efforts.

In addition to optimizing the reporter sequences (**Supplementary Figure 6**), we also tested different divalent metals, temperatures, and salt concentrations to optimize reaction conditions for each nuclease. These data have been added to the revised manuscript (**Supplementary Figs. 2, 4, 7, 12**). Cleavage assays were performed in conditions supporting the highest activity with minimal background. For example, low salt concentration (50 mM) increases rate of reporter cleavage by Can1 and Can2, but results in background nuclease activity in the absence of cA₄ (**Supplementary Fig. 7**). There is certainly more optimization to be done (i.e., the parameter space is infinite), but the work presented here has been a significant investment, and we hope that the reviewer agrees it represents a significant advance that will lead to more nuanced optimization in the future.

4. Related to the point above about the reaction conditions, a sentence like Line 204-205 (comparing performance of TtCan1 and AaCan2) is not very meaningful, unless we know that both enzymes are operating at their optimal conditions. Is 60 deg optimal for Can1, and 55 deg for Can2?

We tested a range of temperatures (45-65°C, in 5°C increments) for each nuclease. These data are now included in the revised manuscript (**Supplementary Figs. 7 & 12**). Optimal temperatures are 50-60°C for TtCan1, 55°C for AaCan2, and 55-60°C for CtNucC.

Are there differences in oligomeric states of Can1 and Can2 which binds to cAs at different molar ratios?

Structures of TtCan1 (PDB:6SCE, McMahon et al., 2020) and SthCan2 (PDB:7BDV; Zhu et al., 2021) were determined in the activator-bound state, while structures of Can2 from *Treponema succinifaciens* (PDB:6WXW; Rostol et al., 2021) were determined with and without cA₄. These studies show that TtCan1, which contains two non-identical CARF domains in one polypeptide, binds cA₄ as a monomer. In contrast, Can2 proteins homodimerize and form a symmetric pocket that accommodates cA₄.

During the revision, we could not reproduce cA₃-dependent activities of Can1 and Can2 nucleases with re-purified protein preparations. Please, see the detailed response to the referee #2 below.

5. Fig 3d: is it possible that NucC can degrade cA4 (though it was previously shown in Gruschow et al to not degrade cA3)? This might explain why supplementing NucC to the Csm/Can2 reaction has a negative effect.

Our biochemical data (**Supplementary Fig. 10f**) shows that the tested NucC orthologs mixed with radiolabeled TtCsm polymerization products do not degrade cA₃ and cA₄. After additional optimization performed to address point 9 below, we re-ran the reactions that combine AaCan2 and CtNucC. This new data shows that combining AaCan2 and CtNucC has no effect, neither positive nor negative, on sensitivity (revised **Fig. 4** and **Supplementary Fig. 13**).

6. Re: the lysis buffer composition assay, it is better to demonstrate this with an AaCan2-based readout instead of the TtCsm6-based readout currently shown. The more sensitive

AaCan2 might be inhibited by different detergents or conditions compared to those affecting Csm6.

We made several significant improvements during the revision, re-tested lysis conditions and repeated assays with patient samples using the updated protocol. This new data is added to the revised manuscript (**Supplementary Fig. 15, Fig. 5**).

7. The authors performed a cross-reactivity assay in their previous publication (Santiago-Frangos et al 2021, which used LAMP in combination with TtCsm) but did not do so here. This is an experiment that should be repeated, as it is not certain the high specificity of detection would be maintained in this LAMP-free, TtCsm/AaCan2-based detection.

This suggestion is more important than we initially thought. The primers used in LAMP or other pre-amplification methods add an upstream selectivity step to the process, which is eliminated during direct detection. We repeated the cross-reactivity assay for Csm-based RNA capture and detection protocol to address the referee's concern. These results are now included in **Fig 5f** and **Supplementary Fig. 17** of the revised manuscript, and we think this is an important addition for all direct detection methods. Cross-reactivity experiments like this are not included in any recent publications on CRISPR-based direct RNA detection (Sridhara S., 2021, Grüşchow S., 2021., Liu T., 2021). These results are an important addition to the current manuscript and will be of broad interest to the field. This new data has been incorporated into the Results and Discussion of the revised manuscript (excerpt included below).

Results, Lines 313-324: "To test cross-reactivity of TtCsm-based RNA capture coupled with AaCan2-based fluorescent detection, we used a panel of seven respiratory viruses, including SARS-CoV-1, Middle East respiratory syndrome coronavirus (MERS-CoV), seasonal coronaviruses 229E (HCoV-229E), NL63 (HCoV-NL63), and HKU1 (HCoV-HKU1). To perform the assay, we used the highest concentration of genomic RNAs available at ATCC and recommended by FDA ($\sim 5 \times 10^5$ copies/ μ L) (<https://www.fda.gov/medical-devices/coronavirus-disease-2019-covid-19-emergency-use-authorizations-medical-devices/in-vitro-diagnostics-euas>). No cross-reactivity is detectable with RNAs from Influenza B, Human respiratory syncytial virus (RSV), and HCoV-229E (**Fig. 5f**). However, we found that reactions with SARS-CoV-1, MERS-CoV, HCoV-HKU1, and HCoV-NL63 RNAs generate a weak signal that is slightly higher than the threshold in negative control samples. Importantly, this signal is significantly lower than the signal obtained for SARS-CoV-2 RNA used at the same concentration ($p < 0.001$, one-way ANOVA with post-hoc Tukey HSD test).

Fig. 5f. Genomic RNAs of common respiratory viruses and coronaviruses related to SARS-CoV-2 (5×10^5 copies/ μ L) were tested with TtCsm-AaCan2 RNA capture and detection assay. Reaction kinetics were measured using a real-time PCR instrument and slopes were quantified in the linear range of the fluorescence curves. Dotted line shows mean of three negative samples (NTC). Mean \pm 2.33 SD for negative samples is shown with gray rectangle.

Discussion, Lines 377-403: *"Direct detection of RNA in clinical samples without RNA extraction or pre-amplification will advance direct-to-consumer diagnostics. However, eliminating pre-amplification removes additional selectivity imparted by the traditional use of primers. To evaluate cross-reactivity of TtCsm-based RNA capture coupled with AaCan2-based fluorescent detection, we tested a panel of seven respiratory viruses. At high RNA concentrations (5×10^5 copies/ μ L) we detect a weak, but reproducible signal to some of the coronaviruses (i.e., SARS-CoV-1, MERS-CoV, HCoV-HKU1, and HCoV-NL63) but not to HCoV-229E or other respiratory viruses (i.e., RSV, Influenza B; **Fig. 5f**). Each of the coronaviruses have 4-7 mismatches in first two segments (S1 - S2) of crRNA:target duplex (**Supplementary Fig. 17**), which permit target RNA binding but significantly decreases polymerization activity of the Cas10 subunit (Steens J., 2021). Coronaviruses that generate a weak signal have two mismatches in the first segment (S1; nucleotides +2 and +5), while HCoV-229E, which generates no signal, has an additional mismatch at the 1st position. The first two nucleotides in segment S1 have the greatest effect on cOA production (Steens, 2021, Nasef M., 2019, Gruschow S., 2021), therefore the additional mismatch in HCoV-229E at the 1st position might explain the complete loss of signal, as compared to the weak signal generated by the other coronaviruses which contain two mismatches in segment 1 (S1). Our original bioinformatic pipeline for crRNA design filtered out guides with potential cross-reactivity, however we prioritized the total number of mismatches in segments S1 and S2 rather than the position of these mismatches (Santiago-Frangos A, 2021). The cross-reactivity assay in this manuscript and published biochemical data (Steens, 2021, Nasef M., 2019, Gruschow S., 2021) demonstrate the importance of mismatch position (e.g., +1 and +2), which is expected to improve the specificity of the next generation of guides. Overall, we anticipate that type III-based RNA pull-down techniques that bypass RNA extraction, combined with further optimization of lysis conditions, more efficient guide design, and the integration of next generation of signal detection methods (e.g., real-time sequencing, digital enzymology, amperometry, etc.) will help bringing type III CRISPR diagnostics closer to current standards of rapid molecular testing.*

8. Line 20 ("we make two major advances that simultaneously limit sample handling and significantly enhance the sensitivity...") is too much of an oversell. The TtCsm/AaCan2 assay still involves multiple liquid handling steps, a bead enrichment step, supernatant removal, resuspension, liquid transfer, etc.

The statement highlighted by the referee compares our protocol for direct detection from swabs to a protocol that requires column-based RNA extraction, isothermal amplification, and type III CRISPR-based fluorescent detection. Further, we streamlined the protocol during the revision, which now includes two steps and a single temperature (see below and in revised **Supplementary Fig. 14 a, c**). We tend to think the collective advancements are "major", but we agree that this term is subjective. The revised abstract no longer uses "major".

9. The high background observed in negative controls is worrisome, and I do not think resorting to reporting slopes is robust enough for clinical diagnosis, especially as multiple liquid/bead handling steps (which can affect individual component concentrations in the reaction, and therefore affect kinetics/slopes) are involved. If the background source is non-sequence specific activation of Cas10 to generate low-level cA4 as the authors suggested (Line 226), is there a way to overcome this?

The quantification approach used here is inspired by other recently published diagnostic methods (*Tina Liu et al, Nature Chem Bio 2021 and Sagar Sridhara et al., Nature Communications 2021*), and the continuity of this approach enables direct comparisons between diagnostic platforms. We recognize the issue related to background activity, and similar background activation was reported in other Type III-based diagnostics (*Sagar Sridhara et al., 2021, Sabine Gruschow et al., 2021*). While Malcolm White's lab demonstrated that additional heparin purification reduces the non-sequence-specific activity of VcCmr-complex (*Sabine Gruschow et al., 2021*), we could not replicate this with the TtCsm-complex. However, reducing TtCsm-complex concentration 50-fold (i.e., 25 nM to 0.5 nM) reduces the background without compromising target-specific activity, which significantly improves the signal-to-noise ($p < 0.001$, revised **Supplementary Fig. 9**) (see below).

10. Related to the point above, the authors might need to do further biochemical investigations on factors that may affect the tool's sensitivity and specificity. Is Cas10 capable of producing enough cA to saturation during the reaction incubation, and what are the rate differences of cA generation from sequence-specific vs non-sequence specific activation? Do the nucleases consume all cAs generated by Cas10?

AaCan2, TtCan1, and NucC effectors used in the viral detection assays in **Figs 3 & 4** have no ring nuclease activities and do not degrade cAs generated by Cas10 subunit of the TtCsm complex after 1h incubation (**Supplementary Figs. 8 and 10f**).

Target-bound TtCsm-complex depletes nearly all the ATP (250 μ M, 1h incubation) in reactions containing the highest target RNA concentrations tested (10^{11} copies/ μ L, ~ 0.17 μ M; **Supplementary Fig 1d**). In reactions performed in the same conditions, but with a 10-fold lower

target concentration, a large fraction of the ATP remains non-polymerized (**Fig 1d**, "*no RNA capture*"). The concentration of SARS-CoV-2 RNA in swab samples ranges approximately from 10^0 to 10^6 copies/ μ L. We know that longer incubation times improve sensitivity (e.g., result in more cA production), but we are trying to develop a rapid detection method, so this has not been our focus. That said, the kinetics of target RNA binding, Cas10 polymerization, cA binding by ancillary nuclease, and kinetics of the ancillary nuclease are all important and are the focus of a new Ph.D. student. This project will be a major effort and will complement but not change the results of the current manuscript. While it is important to know which sensitivities are fundamentally achievable, during the revision we prioritized optimizing enzyme performance (e.g., signal-to-noise, see response to p. 9).

Minor point

- Supplementary Fig 2h has not been referred to in the manuscript (it should be mentioned at around Line 174)

Thank you. A reference to this figure (now **Supplementary Fig. 8a** in the revised text) has been added to the manuscript.

Reviewer #2:

The emergence of the COVID-19 pandemics prompted a fast development of novel diagnostic methods. Class 2 CRISPR-Cas effector complexes as exemplified by Cas12 and Cas13 have been recently harnessed for the molecular diagnostics and development of the point-of-care tests for virus detection. Several attempts were also made to repurpose type III CRISPR systems for the diagnostics of SARS-CoV-2. In 2021, Wiedenheft's lab showed that the Type III effector complex of *Thermus thermophilus* (TtCsm) coupled with the cyclic-oligoadenylate (cA) dependent Csm6 nuclease could be harnessed for the viral RNA detection however detection sensitivity needed to be improved. In this manuscript, Nemudraia et al. sought to increase the sensitivity of TtCsm-based SARS-CoV-2 RNA detection to enable virus detection directly in the patient swab. The authors used two different approaches to improve sensitivity. First, they showed that His-tagged RNase-dead TtCsm bound to viral RNA could be concentrated using nickel-derivatized magnetic beads that results in the increased sensitivity and enables SARS-CoV-2 RNA detection in the nasopharyngeal swab without RNA extraction. However, further optimization was still required to improve sensitivity. Therefore, next Nemudraia et al. probed different cyclic-oligoadenylate dependent nucleases, Can1 and Can2 (cA4-dependent), and NucC (cA3-dependent), for the downstream signal amplification and systematically tested their activation using different cOAs. Unexpectedly, they found that Can1 nuclease from *T. thermophilus* and the Can2 from *Archaeoglobi archaeon* (Aa) are activated both by cA3 and cA4, and that different cAs trigger different substrate specificity (dsDNA or ssRNA, respectively). Since TtCsm complex produces both cA3 and cA4, authors hypothesized that combining cA3- and cA4- sensing nucleases might enhance the sensitivity of TtCsm-based detection. However, coupling of RNase-dead TtCsm complex with two downstream nucleases CtNucC and AaCan2 did not improve the sensitivity compared to the individual nucleases.

Thank you for the thoughtful summary of this paper and how it relates to previous work in the field. As you point out, in the original submission, we reported that Can1/Can2 nucleases are activated both by cA₃ and cA₄. While this dual activity was surprising, it was highly reproducible and led us to speculate about the biological implications of this mechanism. Our revised results demonstrate that TtCan1 and AaCan2 function only as cA₄-activated nucleases (revised **Fig. 2**). Please, see the introduction for a detailed explanation. In response to pts. 1 and 2 raised below, we explain why cA₃ activation of TtCan1 appeared plausible based on available structures and previously published work on Cap4 nucleases activated by asymmetrical signaling molecules (10.1016/j.cell.2020.05.019).

1. Authors propose that various cyclic oligoadenylates, e.g., cA3 and cA4 produced by the TtCsm trigger different nucleic acid specificities providing a fine-tuned mechanism to regulate ancillary nuclease activity in response to the phage infection. However, their conclusion is based on a single point reaction. To support their hypothesis authors have to perform a detailed cleavage analysis of dsDNA, ssDNA or ssRNA to demonstrate clearly which substrate is most favorable for Can1/Can2 nucleases.

Please see the detailed response below (#2).

2. Authors show that Can1 nuclease from *T. thermophilus* and the Can2 from *Archaeoglobi* archaeon are activated both by cA3 and cA4. However, structural studies of Can1, Can2 (and Can2-related nuclease Card1) unambiguously show that binding site for cOA in CARF domain is arranged to accommodate cA4 (but not cA3). Authors should explain how TtCan1 and AaCan2 become activated by an asymmetrical cA3 molecule to degrade dsDNA? What are binding affinities of cA4 and cA3 in this case?

We thank the referee for this question. We were aware of the structure and are familiar with the rules of symmetry, but the results were so robust and reproducible that we found ways to rationalize this unexpected activation. However, questions from the referee's forced us to reconsider this point and prompted us to start fresh with all new reagents. Can1 and Can2 proteins purified from fresh transformants retain cA₄-activated nuclease activity, but neither are activated by cA₃ (**Fig. 2**).

Below is a short summary of observations that supported the cA₃-dependent activities of Can1/Can2. Just to be clear, we are only providing this information to help explain the rationale, but this is not intended to justify our earlier result, which has now been removed from the manuscript.

- A. The crystal structures of Card1 (Can2 homolog) reveals conformational changes in the CARF-domain after binding cA₄ (Rostol et al., 2021). These conformational changes are not seen when Card1 binds cA₆. While cA₄ and cA₆ have a two-fold axis of symmetry, the result reveals binding mechanisms that are amenable to remarkably different ligands.
- B. In the crystal structure of Card1 (PDB:6WXX), cA₄ adopts an “approximately square planar alignment”, but in the TtCan1 (PDB:6SCE) cA₄ has an *asymmetric* confirmation with one of the bases flipped out (see below). Initially, we assumed that the fusion of two CARF domains in TtCan1 might have evolved to bind molecules without two-fold symmetry (i.e., cA₃). A similar mechanism was observed in the SAVED (SMODS-Associated and fused to Various Effector Domains) domain of Cap4 proteins that binds cyclic trinucleotides and was derived from the fusion of two ancestral CARF-like domains (see figure below, 10.1016/j.cell.2020.05.019). In the revised manuscript, we tested additional cyclic nucleotides (i.e., cAG, cA₃, cAAG, etc., **Supplementary Fig 2.**), but cA₄ is the only ligand that activates the nucleases.

3. Authors claim that reproducible cleavage of ssDNA was detected for AaCan2, together with the ssRNase and dsDNase activities (Fig. 2d, Supplementary Fig. 3d). However, data presented in Figure 2d and Supplementary Figure 3d are confusing: ssDNA is cleaved by AaCan2 (Supplementary Figure 3d) but almost no cleavage is observed in gel presented in Figure 2d.

In the previous version of the manuscript two different concentrations of the activator were used for the cleavage assays presented in original **Fig. 2d** (20 nM) and **Supplementary Fig. 3d** (45 nM). Increase of activator concentration leads to a more pronounced cleavage of ssDNA with AaCan2. The two different concentrations were used to highlight an activity that might otherwise go unnoticed.

We re-purified the nucleases and show that this ssDNA activity is reproducible (revised **Supplementary Fig. 4**). Further, we tested additional ssDNA substrate (i.e., phiX174 phage genome) and show that AaCan2 digests this ssDNA in the presence of cA₄ (**Fig. 2h**).

Minor comment:

1) Supplementary Figure 3e: please specify "TtCsm6" above the gel, similar to "AaCan2".

Thank you. We revised the paper according to your suggestion (revised **Supplementary Fig. 8**).

Reviewer #3:

The authors describe a series of experiments to improve the sensitivity and broader applicability of Type III CRISPR diagnostics. Using TtCsm capture and concentrate, the authors improve the sensitivity of Type III diagnostics and enhance this further with the characterization and implementation of accessory proteins (Can1, Can2, NucC). The authors use a nucleic acid biochemistry to profile the Can1 and Can2 accessory nucleases associated with Type III CRISPR-Cas systems. Strikingly, the authors demonstrated that the nature of the secondary messenger cyclic oligo adenylate (C3 or C4) leads to differential nuclease activity. This would suggest that the ligand binding to the CARF domain can allosterically switch the nuclease domain into one of two differing catalytically competent states - an observation that will be of great interest to the field. The manuscript is a valuable contribution to the development of CRISPR diagnostics and contains insights into the biology and mechanism of accessory proteins. Overall, the paper is well written, and the results explained clearly within the text. However, the Figures are somewhat difficult to understand relative to the methods and these should be carefully reviewed to improve clarity before publication.

Thank you for your feedback. Your comments accurately capture our excitement and our surprise.

As explained above, effort to reproduce cA_3 -dependent activation of Can1 and Can2 failed during revision. We provide an extensive explanation above, but in brief, we suspect that our plasmids were contaminated with NucC. The revised manuscript clarifies this issue, and the results demonstrate that TtCan1 and AaCan2 function only as cA_4 -activated nucleases. Please, see detailed descriptions in the short summary in the beginning of the document and in responses to Reviewer #2 (pts. 1 & 2).

We understand the perception of this correction, and deeply regret the mistake. We also understand that this error may change the referee's opinion regarding impact of the work, but ask that you consider new additions to the paper that improve the diagnostic:

- A. We streamlined the procedure of direct RNA detection with type III CRISPR system (from 3 steps to 2 steps)
- B. We decreased the time-to-result (from 33 minutes to 20 minutes)
- C. We improve the signal-to-noise ratio
- D. We added new data related to cross-reactivity

Please see responses to Reviewer#1 for details.

COMMENTS

The authors note an issue with diagnostics sensitivity (specific to Type III systems) and

suggest a capture and concentrate strategy using catalytically deficient TtCsm. Concentrating the RNA using magnetic beads to improve the sensitivity of the diagnostic is an innovative approach but this raises some questions and concerns about specificity. Could the authors comment on how they think capture and concentrate might affect off-target RNA binding and activation of the system? Is the TtCsm complex so specific that it would reject non-target RNA binding? Or is it likely that at very low target concentrations (i.e., those that are clinically relevant) the TtCsm complex may also enrich for off-target RNA (bearing some degree of affinity for the complex) which may in turn raise the chances of an off-target activation of the system?

This is an important question that we had not previously considered in sufficient detail. Reviewer #1 had a similar question (pt. 7 above). In addition to experiments that now measure cross reactivity with seven common respiratory pathogens recommended by FDA (please see response above and new data in **Fig. 5h**), we would also like to point out that the experiments presented here were performed in a complex mixture of human RNAs (total RNA extracted from HEK293T cells or RNA extracted from a patient nasopharyngeal swab) to determine the sensitivity of target RNA detection in the presence of non-target RNAs. This is not always the case for papers that report LoDs for CRISPR-based diagnostics. Reactions with total human RNA alone demonstrate background activation of the complex. However, this background does not significantly differ from a reaction that uses water as an input ($p = 0.98$; **Fig 3d**). This suggests that observed background does not result from off-target binding to human RNA.

FIGURES GENERAL

The authors include a lot of gel images (which is not a problem) but they are cropped and placed side by side where the impression is that they are the same method/stain/label/gel and that one ladder from one gel panel is appropriate or sufficient to interpret the adjacent cropped image. For example, Fig. 2C shows plasmid DNA (agarose) and ssRNA/ssDNA (UREA-PAGE) cleavage gel panels. But it's not clear how these are visualized. Are the nucleic acids tagged? Was the gel post stained? The authors are strongly encouraged to consider including schematics or appropriate text in the Figure or legend to help the reader understand what the experiment is showing.

We apologize for the confusion. In the revised manuscript we repeated all cleavage assays with new preparations of Can1/Can2 nucleases and re-ran all gels with appropriate MW ladders. Corresponding labels and figure legends are revised to clarify how the assays were performed and visualized.

Fig. 2B: The phylogenetic tree has no scale and needs one. Same for Supplementary Fig. 6A

Thank you for pointing this out. Scale bars have been added.

REVIEWERS' COMMENTS

Reviewer #1 (Remarks to the Author):

Overall I am satisfied with the revision. I appreciate the authors' honesty and careful work in figuring out the problem behind the intriguing cA3-dependent activation of Can1/2. Despite this observation not being reproducible, the authors provided many useful observations, including sequence preference for ancillary nucleases useful for selecting a reporter type/sequence; optimal conditions for nuclease functions; and suggestions to improve the specificity of the detection system. These info could enable researchers in the field to further develop type III CRISPR systems into robust diagnostic tools.

Reviewer #2 (Remarks to the Author):

I went through the revised manuscript. Formally, the authors addressed all my concerns but this led to a major revision of the manuscript and the removal of cA3-stimulated activities of AaCan2 from the manuscript. I am not sure whether it is suitable for Nature Communications since it now mainly provides technical improvements of the original technology that are significant but the scientific novelty is limited.

Reviewer Reports on the Second Revision:

Reviewer #1 (Remarks to the Author):

Overall I am satisfied with the revision. I appreciate the authors' honesty and careful work in figuring out the problem behind the intriguing cA3-dependent activation of Can1/2. Despite this observation not being reproducible, the authors provided many useful observations, including sequence preference for ancillary nucleases useful for selecting a reporter type/sequence; optimal conditions for nuclease functions; and suggestions to improve the specificity of the detection system. These info could enable researchers in the field to further develop type III CRISPR systems into robust diagnostic tools.

Reviewer #2 (Remarks to the Author):

I went through the revised manuscript. Formally, the authors addressed all my concerns but this led to a major revision of the manuscript and the removal of cA3-stimulated activities of AaCan2 from the manuscript. I am not sure whether it is suitable for Nature Communications since it now mainly provides technical improvements of the original technology that are significant but the scientific novelty is limited.

Author Rebuttals to Second Revision:

We thank the reviewers again for the careful review of our work.